# High-Dimensional Gaussian Process Regression with Soft Kernel Interpolation

**Chris Camaño**                                           *ccamano@caltech.edu*
*Department of Computing and Mathematical Sciences,*
*California Institute of Technology, Pasadena, CA, 91125, USA*

**Daniel Huang**                                           *dan@base26labs.com*
*Base26, CA, USA*

**Reviewed on OpenReview:** *https://openreview.net/forum?id=U9b2FIjvWU*

## Abstract

We introduce *Soft Kernel Interpolation* (SoftKI), a method that combines aspects of Structured Kernel Interpolation (SKI) and variational inducing point methods, to achieve scalable Gaussian Process (GP) regression on high-dimensional datasets. SoftKI approximates a kernel via softmax interpolation from a smaller number of interpolation points learned by optimizing a combination of the SoftKI marginal log-likelihood (MLL), and when needed, an approximate MLL for improved numerical stability. Consequently, it can overcome the dimensionality scaling challenges that SKI faces when interpolating from a dense and static lattice while retaining the flexibility of variational methods to adapt inducing points to the dataset. We demonstrate the effectiveness of SoftKI across various examples and show that it is competitive with other approximated GP methods when the data dimensionality is modest (around 10).

## 1 Introduction

Gaussian processes (GPs) are flexible function approximators based on Bayesian inference. However, there are scaling concerns. For a dataset comprising $n$ data points, constructing the $n \times n$ kernel (covariance) matrix required for GP inference incurs a space complexity of $\mathcal{O}(n^2)$. Naively, posterior inference using direct solvers costs $\mathcal{O}(n^3)$ time, which quickly becomes infeasible for moderate $n$. In recent years, this shortcoming has led to the widespread adoption of conjugate gradient (CG) based GPs enabling "exact" GP inference in $\mathcal{O}(n^2)$ time (Cunningham et al., 2008; Cutajar et al., 2016). Despite these developments, exact GP regression remains challenging in practice, often requiring bespoke engineering solutions (Wang et al., 2019; Gardner et al., 2018a).

A popular response to this scaling issue is based on *variational inference methods* (Titsias, 2009; Hensman et al., 2013). These approaches build a variational approximation of the posterior GP by learning the locations of $m \ll n$ *inducing points* (Quinonero-Candela & Rasmussen, 2005; Snelson & Ghahramani, 2005). Inducing points and their corresponding *inducing variables* introduce latent variables with normal priors that form a low-rank approximation of the covariance structure in the original dataset. This approach improves the time complexity of posterior inference to $\mathcal{O}(nm^2)$ for Sparse Gaussian Process Regression (SGPR) (Titsias, 2009) and $\mathcal{O}(m^3)$ for Stochastic Variational Gaussian Process Regression (SVGP) (Hensman et al., 2013), with the latter introducing additional variational parameters.

Another successful approach is based on *Structured Kernel Interpolation* (SKI) (Wilson & Nickisch, 2015) and its variants such as product kernel interpolation (SKIP) (Gardner et al., 2018b), Simplex-SKI (Kapoor et al., 2021), and Sparse-Grid SKI (Yadav et al., 2022). SKI-based methods achieve scalability by constructing computationally tractable approximate kernels via interpolation from a pre-computed and dense rectilinear grid of *interpolation points*. This structure enables fast matrix-vector multiplications (MVMs), which leads to

downstream acceleration when paired with CG-based kernel inversion strategies. Superficially, this approach effectively responds to the aforementioned scaling issues, but unfortunately has the consequence of causing the complexity of posterior inference to become explicitly dependent on the dimensionality $d$ of the data (*e.g.*, $\mathcal{O}(n4^d + dm^d \log m)$ for a MVM in SKI and $\mathcal{O}(d^2(n+m))$ for a MVM in Simplex-SKI). Moreover, the static grid used in SKI approximations does not have the flexibility of a variational method to adapt to the dataset at hand. This limitation motivates us to search for accurate and scalable GP regression algorithms that can attain the best of both approaches.

In this paper, we introduce *soft kernel interpolation* (SoftKI), which combines aspects of inducing points and SKI to enable scalable GP regression on high-dimensional datasets. Our main observation is that while SKI can support large numbers of interpolation points, it uses them in a sparse manner—only a few interpolation points contribute to the interpolated value of any single data point. Consequently, we should still be able to interpolate successfully provided that we place enough of them in good locations relative to the dataset, *i.e.*, learn their locations as in a variational approach. This change removes the explicit dependence of the cost of the method on the data dimensionality $d$. It also opens the door to more directly optimize with an approximate GP marginal log-likelihood (MLL) to learn the locations of interpolations points as opposed to a variational approximation that introduces latent variables with normal priors over the values observed at inducing point locations.

Our method approximates a kernel by interpolation from a softmax of $m \ll n$ learned interpolation points, hence the name *soft kernel interpolation* (Section 3.1). The interpolation points are learned by optimizing a combination of the SoftKI MLL, and when needed, an approximate MLL for improved numerical stability in single-precision floating point arithmetic (Section 3.2). It is able to leverage GPU acceleration and stochastic optimization for scalability. Since the kernel structure is dynamic, we rely on $m \ll n$ interpolation points to obtain a time complexity of $\mathcal{O}(nm^2)$ and space complexity of $\mathcal{O}(nm)$ for posterior inference (Section 3.3). We evaluate SoftKI on a variety of datasets from the `UCI` repository (Kelly et al., 2017) and demonstrate that it achieves test root mean square error (RMSE) comparable to other inducing point methods for datasets with moderate dimensionality (approximately $d = 10$) (Section 4.1). To further explore the effectiveness of SoftKI in high dimensions, we apply SoftKI to molecule datasets ($d$ in the hundreds to thousands) and show that SoftKI is scalable and competitive in these settings as well (Section 4.2). Lastly, we examine the numerical stability of SoftKI (Section 4.3).

## 2 Background and Related Work

**Notation.** Matrices will be denoted by upper-case boldface letters (e.g. $\mathbf{A} \in \mathbb{R}^{n \times m}$). When specifying a matrix entry-wise generated by an underlying function we write $\mathbf{A} = [g(i,j)]_{i,j}$, meaning $\mathbf{A}_{ij} = g(i,j)$. At times, given points $x_1, \ldots, x_n \in \mathbb{R}^d$, we stack them into $\mathbf{x} \in \mathbb{R}^{nd}$ by $\mathbf{x}^\top = (x_1^\top, \ldots, x_n^\top)$, so that entries $d(i-1) + 1$ through $di$ are the components of $\mathbf{x}_i$. Any function $f : \mathbb{R}^d \to \mathbb{R}$ extends column-wise via $f(\mathbf{x}) = (f(\mathbf{x}_1), \ldots, f(\mathbf{x}_n))^\top$.

### 2.1 Gaussian Processes

Let $k : \mathbb{R}^d \times \mathbb{R}^d \to \mathbb{R}$ be a positive semi-definite kernel function. A (centered) *Gaussian process* (GP) with kernel $k$ is a distribution over functions $f : \mathbb{R}^d \to \mathbb{R}$ such that for any finite collection of inputs $\{x_i\}_{i=1}^n$, the vector of function values $f(\mathbf{x})$ is jointly Gaussian $f(\mathbf{x}) \sim \mathcal{N}(\mathbf{0}, \mathbf{K_{xx}})$, where the kernel matrix $\mathbf{K_{xx}} \in \mathbb{R}^{n \times n}$ has entries $[\mathbf{K_{xx}}]_{ij} = k(x_i, x_j)$. Given two collections $\{x_i\}_{i=1}^n$ and $\{x'_j\}_{j=1}^{n'}$, the cross-covariance matrix is denoted $\mathbf{K_{xx'}} \in \mathbb{R}^{n \times n'}$ is defined by $[\mathbf{K_{xx'}}]_{ij} = k(x_i, x'_j)$.

To perform GP regression on the labeled dataset $\mathcal{D} := \{(x_i, y_i) : x_i \in \mathbb{R}^d, y_i \in \mathbb{R}\}_{i=1}^n$, we assume the data is generated as follows:

$$f(\mathbf{x}) \sim \mathcal{N}(\mathbf{0}, \mathbf{K_{xx}}) \tag{GP}$$

$$\mathbf{y} \,|\, f(\mathbf{x}) \sim \mathcal{N}(f(\mathbf{x}), \beta^2 \mathbf{I}) \tag{likelihood}$$

where $f$ is a function sampled from a GP and each observation $y_i$ is $f$ evaluated at $x_i$ perturbed by independent and identically distributed (i.i.d.) Gaussian noise with zero mean and variance $\beta^2$. The posterior

predictive distribution has the following closed-form solution (Rasmussen & Williams, 2005)

$$p(f(*) \mid \mathbf{x}, \mathbf{y}) = \mathcal{N}(\mathbf{K}_{*\mathbf{x}}(\mathbf{K}_{\mathbf{xx}} + \mathbf{\Lambda})^{-1}\mathbf{y}, \mathbf{K}_{**} - \mathbf{K}_{*\mathbf{x}}(\mathbf{K}_{\mathbf{xx}} + \mathbf{\Lambda})^{-1}\mathbf{K}_{\mathbf{x}*}) \tag{1}$$

where $\mathbf{\Lambda} = \beta^2 \mathbf{I}$. Using direct methods, the time complexity of inference is $\mathcal{O}(n^3)$ which is the complexity of solving the system of linear equations in $n$ variables $(\mathbf{K}_{\mathbf{xx}} + \mathbf{\Lambda})\alpha = \mathbf{y}$ for $\alpha$ so that the posterior mean (Equation 1) is $\mathbf{K}_{*\mathbf{x}}\alpha$.

A GP's *hyperparameters* $\theta$ include the noise $\beta^2$ and other parameters such as those involved in the definition of a kernel such as its *lengthscale* $\ell$ and *output scale* $\sigma$. Thus $\theta = (\beta, \ell, \sigma)$. The hyperparameters can be learned by maximizing the MLL of a GP

$$\log p(\mathbf{y} \mid \mathbf{x}; \theta) = \log \mathcal{N}(\mathbf{y} \mid \mathbf{0}, \mathbf{K}_{\mathbf{xx}}(\theta) + \mathbf{\Lambda}(\theta))$$

where $\mathcal{N}(\cdot \mid \mu, \mathbf{\Sigma})$ is notation for the probability density function (PDF) of a Gaussian distribution with mean $\mu$ and covariance $\mathbf{\Sigma}$ and we have explicitly indicated the dependence of $\mathbf{K}_{\mathbf{xx}}$ and $\mathbf{\Lambda}$ on $\theta$.

## 2.2 Sparse Gaussian Process Regression

Many scalable GP methods address the high computational cost of GP inference by approximating the kernel using a Nyström method (Williams & Seeger, 2000). This approach involves selecting a smaller set of $m$ inducing points, $\mathbf{z} \subset \mathbf{x}$, to serve as representatives for the complete dataset. The original $n \times n$ kernel $\mathbf{K}_{\mathbf{xx}}$ is then approximated as

$$\mathbf{K}_{\mathbf{xx}}^{\mathrm{SGPR}} = \mathbf{K}_{\mathbf{xz}}\mathbf{K}_{\mathbf{zz}}^{-1}\mathbf{K}_{\mathbf{zx}} \approx \mathbf{K}_{\mathbf{xx}},$$

leading to an overall inference complexity of $\mathcal{O}(nm^2)$.

Variational inducing point methods such as SGPR (Titsias, 2009) combines the Nyström GP kernel approximation with a variational optimization procedure so that the positions of the inducing points can be learned. The variational approximation introduces latent *inducing variables* $f(\mathbf{z})$ to model the values observed at inducing points $\mathbf{z}$ with joint distribution

$$p(f(\mathbf{x}), f(\mathbf{z})) = \mathcal{N}\left(\begin{pmatrix} f(\mathbf{x}) \\ f(\mathbf{z}) \end{pmatrix} \middle| \begin{pmatrix} \mathbf{0} \\ \mathbf{0} \end{pmatrix}, \begin{pmatrix} \mathbf{K}_{\mathbf{xx}} & \mathbf{K}_{\mathbf{xz}} \\ \mathbf{K}_{\mathbf{zx}} & \mathbf{K}_{\mathbf{zz}} \end{pmatrix}\right).$$

The inducing points are learned by maximizing the *evidence lower bound* (ELBO)

$$\mathrm{ELBO}(q) = \mathbb{E}_{q(\mathbf{Z})}\big[\log(p(\mathbf{Y} \mid \mathbf{Z})\big] - \mathrm{KL}(q(\mathbf{Z}) \,\|\, p(\mathbf{Z})) \tag{2}$$

where KL is the KL divergence between two probability distributions, $q$ is a computationally tractable variational distribution, $\mathbf{Y}$ is observed data ($\mathbf{Y} = \mathbf{y}$ for SGPR), and $\mathbf{Z}$ are latent variables ($\mathbf{Z} = (f(\mathbf{x}), f(\mathbf{z}))$ for SGPR). In the case of SGPR regression, the variational family $q(f(\mathbf{x}), f(\mathbf{z})) = p(f(\mathbf{x}) \mid f(\mathbf{z}))q(f(\mathbf{z}))$ is chosen to approximate $p(f(\mathbf{x}), f(\mathbf{z}))$ so that Equation 2 simplifies to

$$\mathrm{ELBO}(q) = \log \mathcal{N}\left(\mathbf{y} \mid \mathbf{0}, \mathbf{K}_{\mathbf{xx}}^{\mathrm{SGPR}} + \mathbf{\Lambda}\right) - \frac{1}{2}\,\mathrm{tr}\left(\mathbf{K}_{\mathbf{xx}} - \mathbf{K}_{\mathbf{xx}}^{\mathrm{SGPR}}\right).$$

Since the ELBO is a lower bound on the MLL $\log p(\mathbf{y} \mid \mathbf{x}; \theta)$, we can maximize the ELBO via gradient-based optimization as a proxy for maximizing $\log p(\mathbf{y} \mid \mathbf{x}; \theta)$ to learn the location of the inducing points $\mathbf{z}$ by treating it as a GP hyperparameter, *i.e.*, $\theta = (\beta, \ell, \sigma, \mathbf{z})$. The time complexity of computing the ELBO is $\mathcal{O}(nm^2)$.

The posterior predictive distribution has closed form

$$q(f(*) \mid \mathbf{x}, \mathbf{y}) = \mathcal{N}(f(*) \mid \mathbf{K}_{*\mathbf{z}}\mathbf{C}^{-1}\mathbf{K}_{\mathbf{zx}}\mathbf{\Lambda}^{-1}\mathbf{y}, \mathbf{K}_{**} - \mathbf{K}_{*\mathbf{z}}(\mathbf{K}_{\mathbf{zz}}^{-1} - \mathbf{C}^{-1})\mathbf{K}_{\mathbf{z}*})$$

where $\mathbf{C} = \mathbf{K}_{\mathbf{zz}} + \mathbf{K}_{\mathbf{zx}}\mathbf{\Lambda}^{-1}\mathbf{K}_{\mathbf{xz}}$. The time complexity of posterior inference is $\mathcal{O}(nm^2 + m^3)$ since it requires solving $\mathbf{C}\alpha = \mathbf{K}_{\mathbf{zx}}\mathbf{\Lambda}^{-1}\mathbf{y}$ for $\alpha$, which is dominated by the cost of forming $\mathbf{C}$.

**Additional variational methods.** Building on SGPR, SVGP (Hensman et al., 2013) extends the optimization process used for SGPR to use stochastic variational inference. This further reduces the computational

cost of GP inference to $\mathcal{O}(m^3)$ since optimization is now over a variational distribution on $\mathbf{z}$, rather than the full posterior. Importantly SVGP is highly scalable to GPUs since its optimization strategy is amenable to minibatch optimization procedures. More recent research has introduced Variational Nearest Neighbor Gaussian Processes (VNNGP) (Wu et al., 2022), which replaces the low-rank prior of SVGP with a sparse approximation of the precision matrix by retaining only correlations among each point's $K$ nearest neighbors. By constructing a sparse Cholesky factor with at most $K + 1$ nonzeros per row, VNNGP can reduce the per-iteration cost of evaluating the SVGP objective (Equation 2.2) to $\mathcal{O}\big((n_b + m_b)K^3\big)$ when minibatching over $n_b$ data points and $m_b$ inducing points.

## 2.3 Structured Kernel Interpolation

SKI approaches scalable GP regression by approximating a large covariance matrix $\mathbf{K_{xx}}$ by interpolating from cleverly chosen interpolation points. In particular, SKI makes the approximation

$$\mathbf{K}_{\mathbf{xx'}}^{\mathrm{SKI}} = \mathbf{W_{xz}} \, \mathbf{K_{zz}} \, \mathbf{W_{zx}} \approx \mathbf{K_{xx}}.$$

where $\mathbf{W_{xz}}$ is a sparse matrix of interpolation weights, $\mathbf{W_{zx}} = \mathbf{W_{xz}^\top}$, and $\mathbf{z}$ are interpolation points. The interpolation points can be interpreted as quasi-inducing points since

$$\mathbf{K}_{\mathbf{xx'}}^{\mathrm{SKI}} = (\mathbf{W_{xz}K_{zz}}) \, \mathbf{K_{zz}^{-1}} \, (\mathbf{K_{zz}W_{zx}}) = \widehat{\mathbf{K}}_{\mathbf{xz}} \, \mathbf{K_{zz}^{-1}} \, \widehat{\mathbf{K}}_{\mathbf{zx}}.$$

where $\widehat{\mathbf{K}}_{\mathbf{xz}} = \mathbf{W_{xz}K_{zz}}$. However, interpolation points are *not* associated with inducing variables since they are used for kernel interpolation rather than defining a variational approximation. The posterior predictive distribution is then the standard GP posterior predictive

$$p(f(*) \,|\, \mathbf{x}, \mathbf{y}) \approx \mathcal{N}(\mathbf{K_{*x}}(\mathbf{K_{xx}^{SKI}} + \mathbf{\Lambda})^{-1}\mathbf{y}, \mathbf{K_{**}} - \mathbf{K_{*x}}(\mathbf{K_{xx}^{SKI}} + \mathbf{\Lambda})^{-1}\mathbf{K_{x*}}) \tag{3}$$

where $\mathbf{K_{xx}}$ is replaced with $\mathbf{K_{xx}^{SKI}}$. When Equation 3 is solved via linear CG with structured interpolation (*e.g.*, cubic), the resulting method is called KISS-GP (Wilson & Nickisch, 2015). Since the time complexity of CG depends on access to fast MVMs, KISS-GP makes two design choices that are compatible with CG. First, KISS-GP uses cubic interpolation weights (Keys, 1981) so that $\mathbf{W_{xz}}$ has only four non-zero entries in each row, i.e., $\mathbf{W_{xz}}$ is sparse. Second, KISS-GP chooses the interpolation points $\mathbf{z}$ to be on a fixed rectilinear grid in $\mathbb{R}^d$ which induces multilevel Toeplitz structure in $\mathbf{K_{zz}}$ if $k(x, y)$ is a stationary kernel.

Putting the two together, the resulting kernel $\mathbf{K_{xx}^{SKI}}$ is a highly structured structured matrix which enables fast MVMs. If we wish to have $m$ distinct interpolation points for each dimension, the grid will contain $m^d$ interpolation points leading to a total complexity of GP inference of $\mathcal{O}(n4^d + dm^d \log(m))$ since there are $\mathcal{O}(4^d)$ nonzero entries per row of $\mathbf{W_{xz}}$ and a MVM with a Toeplitz structured kernel like $\mathbf{K_{zz}}$ can be done in $\mathcal{O}(|\mathbf{z}| \log(|\mathbf{z}|))$ time via a Fast Fourier Transform (FFT) (Wilson, 2014). Thus, for low dimensional problems, SKI can provide significant speedups compared to vanilla GP inference and supports a larger number of interpolation points compared to SGPR.

## 2.4 Other Variants of Structured Kernel Interpolation.

While SKI can provide significant speedups over vanilla GP inference, its scaling in $d$ restricts its usage to low-dimensional settings where $4^d < n$. To address this, Gardner et al. (2018b) introduced SKIP, which optimizes SKI by expressing a $d$-dimensional kernel as a product of $d$ one-dimensional kernels. This reduces the MVM cost with $\mathbf{K_{zz}}$ from $\mathcal{O}(dm^d \log(m))$ to $\mathcal{O}(dm \log(m))$, using only $m$ grid points per component kernel instead of $m^d$. However, SKIP is limited to dimensions roughly in the range of $d = 10$–$30$ for large datasets, as it requires substantial memory and may suffer from low-rank approximation errors.

Recent work on improving the scaling properties of SKI has been focused on efficient incorporation of simplicial interpolation, which when implemented carefully in the case of Simplex-GP (Kapoor et al., 2021) and Sparse-Grid GP (Yadav et al., 2022) can reduce the cost of a single MVM with $\mathbf{W_{xz}}$ down to $\mathcal{O}\left(nd^2\right)$. These methods differ in the grid construction used for interpolation with Simplex-SKI opting for a permutohedral lattice (Adams et al., 2010), while Sparse-Grid GP uses sparse grids (Bungartz & Griebel, 2004) and a custom recursive MVM algorithm. These modifications vastly improve KISS-GP leading to $\mathbf{K_{zz}}$ MVM costs of

Table 1: Cost of a single MVM with the approximate covariance for various GP approximations. For listed methods based on SKI we take $m$ points in each dimension. The variable $r$ represents the rank used in the SKIP approximation ($r \approx 100$), and the variable $\ell$ is the *grid resolution* taken in Sparse-Grid GP (typically $\ell \approx 1-5$)

| Method | Time Complexity of one MVM |
|---|---|
| KISS-GP (cubic) | $\mathcal{O}\big(n4^d + dm^d \log(m)\big)$ |
| SKIP | $\mathcal{O}\big(d(rn + m \log m)\big)$ |
| Simplex-GP | $\mathcal{O}\big(d^2(n + m)\big)$ |
| Sparse-Grid GP (simplicial) | $\mathcal{O}\big(nd^{2+\ell} + \ell^d 2^\ell\big)$ |
| SGPR | $\mathcal{O}\big(nm^2 + m^3\big)$ |
| SoftKI | $\mathcal{O}\big(nm^2 + m^3\big)$ |

$\mathcal{O}(d^2(n+m))$ for Simplex-GP and $\mathcal{O}\big(nd^{2+\ell} + \ell^d 2^\ell\big)$ for Sparse-Grid GP where $\ell$ is a small constant. While both of these methods greatly improve the scaling of SKI, they still feature a problematic dependency on $d$ limiting their extension to arbitrarily high dimensional datasets. The complexity of SKI and related variants are tabulated in Table 1.

## 3 Soft Kernel Interpolation

In this section, we introduce SoftKI (Algorithm 1). SoftKI takes the same starting point as SKI, namely that $\mathbf{K}_{\mathbf{xx}}^{\mathrm{SKI}} = \mathbf{W}_{\mathbf{xz}}\mathbf{K}_{\mathbf{zz}}\mathbf{W}_{\mathbf{zx}}$ can be used as an approximation of $\mathbf{K}_{\mathbf{xx}}$. However, we will deviate in that we will abandon the structure given by a lattice and opt to learn the locations of the interpolation points $\mathbf{z}$ instead. This raises several issues. First, we revisit the choice of interpolation scheme since we no longer assume a static structure (Section 3.1). Second, we require an efficient procedure for learning the interpolation points (Section 3.2). Third, we require an alternative route to recover a scalable GP since our approach removes structure in the kernel that was used in SKI for efficient inference (Section 3.3).

### 3.1 Soft Interpolation Strategies

Whereas a cubic (or higher-order) interpolation scheme is natural in SKI when using a static lattice structure, our situation is different since the locations of the interpolation points are now dynamic. In particular, we would ideally want our interpolation scheme to have the flexibility to adjust the contributions of interpolation points used to interpolate a kernel value in a data-dependent manner. To accomplish this, we propose *softmax interpolation weights.*

**Definition 3.1** (Softmax interpolation weights). For a finite collection of inputs $\{x_i\}_{i=1}^n$, $x_i \in \mathbb{R}^d$, and a set of interpolation points $\{z_j\}_{j=1}^m$, $z_j \in \mathbb{R}^d$, define the *softmax interpolation weight matrix* as

$$\mathbf{\Sigma}_{\mathbf{xz}} = \left[ \frac{\exp\left(-\|x_i - z_j\|\right)}{\sum_{k=1}^m \exp\left(-\|x_i - z_k\|\right)} \right]_{ij}. \tag{4}$$

With this definition in hand we can now define the SoftKI kernel approximation

$$\mathbf{K}_{\mathbf{xx}}^{\mathrm{SoftKI}} = \mathbf{\Sigma}_{\mathbf{xz}}\mathbf{K}_{\mathbf{zz}}\mathbf{\Sigma}_{\mathbf{zx}} \approx \mathbf{K}_{\mathbf{xx}}.$$

Since a softmax can be interpreted as a probability distribution, we can view SoftKI as interpolating from a probabilistic mixture of interpolation points. Each input point $x_i$ is interpolated from all points, with an exponential weighting favoring closer interpolation points (See Figure 1). Each row $\mathbf{\Sigma}_{x_i \mathbf{z}}$ represents the softmax interpolation weights between a single point $x_i$ and all $m$ interpolation points. Note that $\mathbf{\Sigma}_{\mathbf{xz}}$ is not strictly sparse since softmax interpolation continuously assigns weights over each interpolation point for each $x_i$. However, as $d$ increases, the weights for distant points become negligible. This leads to an

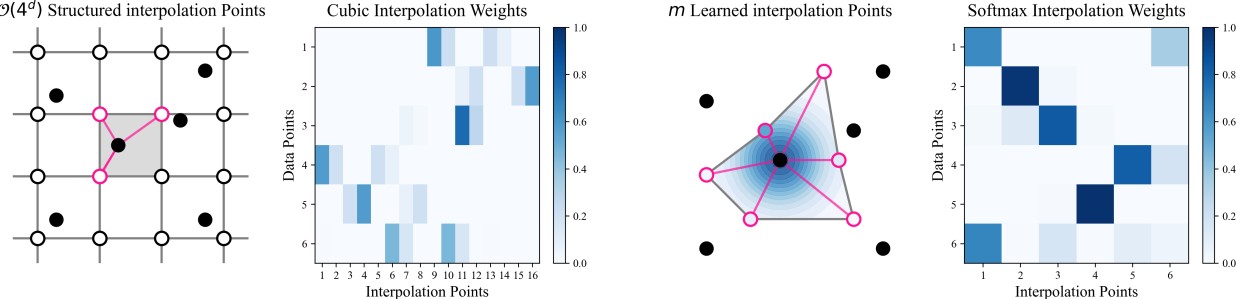

Figure 1: **Structured & Soft Kernel Interpolation:** Comparison of interpolation procedure for local cubic interpolation during KISS-GP (*left*) and softmax interpolation (*right)* during SoftKI. Here white points ○ indicate interpolation points $z_i \in \mathbf{z}$ and black points ● are data points $x_i \in \mathbf{x}$. In these diagrams magenta line segments indicate which interpolation points are being used for a given data point during the interpolation procedure of each method. In KISS-GP, a small local subset of grid points is used, whereas in SoftKI all interpolation points are included, weighted by a softmax distribution centered at a given data point.

effective sparsity, with each $x_i$ predominantly influenced only by its nearest interpolation points during the reconstruction of $\mathbf{K_{xx}}$.

To add more flexibility to the model, the softmax interpolation scheme can be extended to contain a learnable temperature parameter $T$ as below

$$\mathbf{\Sigma_{xz}} = \left[ \frac{\exp\left(-\|x_i/T - z_j\|\right)}{\sum_{k=1}^{m} \exp\left(-\|x_i/T - z_k\|\right)} \right]_{ij}.$$

This additional hyperparameter is needed because unlike the SGPR kernel $\mathbf{K_{xx}^{SGPR}} = \mathbf{K_{xz}K_{zz}^{-1}K_{zx}}$ where the lengthscale influences each term, the lengthscale only affects $\mathbf{K_{zz}}$ in the SoftKI kernel. The temperature acts as lengthscale-like hyperparameter on the interpolation scheme that controls the distance between the datapoints $\mathbf{x}$ and the interpolation points $\mathbf{z}$. When $T = 1$, we obtain the original scheme give in Equation 4. Analogous to how automatic relevance detection (ARD) (MacKay et al., 1994) can be used to set lengthscales for different dimensions, we can also use a different temperature per dimension. When learning both temperatures and lengthscales, we cap the lengthscale range for more stable hyperparameter optimization since they have a push-and-pull effect.

### 3.2 Learning Interpolation Points

Similar to a SGPR, we can treat the interpolation points $\mathbf{z}$ as hyperparameters of a GP and learn them by optimizing an appropriate objective. We propose optimizing the MLL $\log p(\mathbf{y} \mid \mathbf{x}; \theta)$ with a method based on gradient descent for a SoftKI which for $\mathbf{D}_\theta = \mathbf{K_{xx}^{SoftKI}}(\theta) + \mathbf{\Lambda}(\theta)$ has closed the form solution

$$\log p(\mathbf{y} \mid \mathbf{x}; \theta) = -\frac{1}{2}\left[\mathbf{y}^\top \mathbf{D}_\theta^{-1}\mathbf{y} + \log\det(\mathbf{D}_\theta) + n\log(2\pi)\right], \tag{5}$$

and derivative

$$\frac{\partial \log p(\mathbf{y} \mid \mathbf{x}; \theta)}{\partial \theta} = -\frac{1}{2}\left[\mathbf{y}^\top \mathbf{D}_\theta^{-1}\frac{\partial \mathbf{D}_\theta}{\partial \theta}\mathbf{D}_\theta^{-1}\mathbf{y} + \text{tr}\left(\mathbf{D}_\theta^{-1}\frac{\partial \mathbf{D}_\theta}{\partial \theta}\right)\right]. \tag{6}$$

Each evaluation of $\log p(\mathbf{y} \mid \mathbf{x}; \theta)$ has time complexity $\mathcal{O}(nm^2)$ and space complexity $\mathcal{O}(nm)$ where $m$ is the number of interpolation points.

**Hutchinson's pseudoloss.** Occasionally, we observe that the MLL is numerically unstable in single-precision floating point arithmetic (see Section 4.3 for further details). To help stabilize the MLL in these

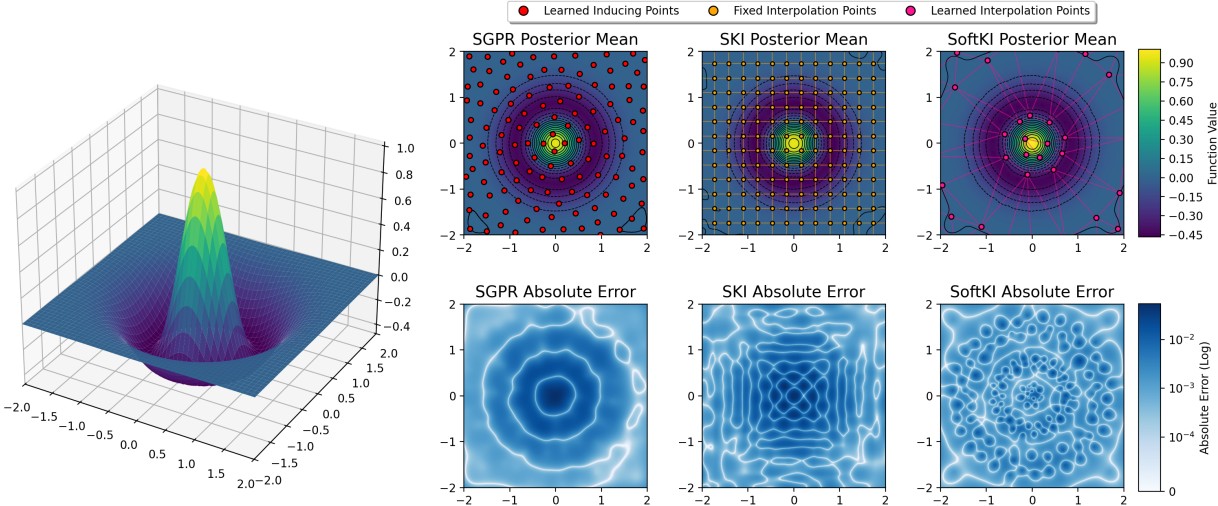

Figure 2: **Inducing vs. Interpolation Points:** Comparison of inducing points learned by SGPR, static rectilinear lattice points in SKI, and adaptive interpolation points learned by SoftKI on the Ricker wavelet (Ricker, 1953) after 100 epochs of hyperparameter optimization. Contour plots of the posterior mean are paired with each method's absolute-error. Each method achieves comparable accuracy: SGPR's inducing points follow the sample distribution, and SoftKI's interpolation points adapt to the true function's local geometry. Additional experiment details can be found in Appendix C.1

situations, we use an approximate MLL based on work in approximate GP theory (Gardner et al., 2018b; Maddox et al., 2021; Wenger et al., 2023) that identify efficiently computable estimates of the log determinant (Equation 5) and trace term (Equation 6) in the GP MLL. In more detail, these methods combine stochastic trace estimation via the Hutchinson's trace estimator (Girard, 1989; Hutchinson, 1989) with blocked CG so that the overall cost remains quadratic in the size of the kernel. Maddox et al. (2022) term this approximate MLL the *Hutchinson's pseudoloss* and show that it remains stable under low-precision conditions.

**Definition 3.2** (Hutchinson Pseudoloss). Let $\mathbf{u}_0, \mathbf{u}_1, \ldots, \mathbf{u}_l$ be the solutions obtained by using block conjugate gradients for the system $\mathbf{D}_\theta(\mathbf{u}_0\ \mathbf{u}_1 \ldots \mathbf{u}_l) = (\mathbf{y}\ \mathbf{w}_1 \ldots \mathbf{w}_l)$, where each $\mathbf{w}_j \in \mathbb{R}^n$ is a Gaussian random vector normalized to unit length. The Hutchinson pseudoloss approximation of the GP marginal log likelihood in Equation 5 is given as

$$\log \tilde{p}(\mathbf{y} \mid \mathbf{x}; \theta) = -\frac{1}{2}\left[\mathbf{u}_0^\top \mathbf{D}_\theta \mathbf{u}_0 + \frac{1}{l}\sum_{j=1}^{l} \mathbf{u}_j^\top (\mathbf{D}_\theta \mathbf{w}_j)\right] \tag{7}$$

with derivative

$$\frac{\partial \log \tilde{p}(\mathbf{y} \mid \mathbf{x}; \theta)}{\partial \theta} = -\frac{1}{2}\left[\mathbf{u}_0^\top \frac{\partial \mathbf{D}_\theta}{\partial \theta}\mathbf{u}_0 + \frac{1}{l}\sum_{j=1}^{l} \mathbf{u}_j^\top \frac{\partial \mathbf{D}_\theta}{\partial \theta}\mathbf{w}_j\right] \tag{8}$$

By computing Equation 7 as described above as opposed to explicitly computing a stochastic Lanczos quadrature approximation (Ubaru et al., 2017), the gradient can also be efficiently computed using back propagation. Note that the gradient of Hutchinson's pseudoloss approximates the gradient of the GP MLL, where the trace term is estimated using a stochastic trace estimator. The accuracy of this approximation depends on the number of probe vectors $\ell$. Consequently, gradient-based optimization of the GP MLL can be approximated by optimizing Hutchinson's pseudoloss instead.

---

**Algorithm 1** SoftKI Regression. The procedure `kmeans` performs k-means clustering, `batch` splits the dataset into batches, and `softmax_interpolation` produces a softmax interpolation matrix (see Section 3.1).

---

**Require:** SoftKI GP hyperparameters $\theta = (\beta, \ell, \sigma, \mathbf{z} \in \mathbb{R}^{(m \times d)}, T)$, kernel function $k(x, y)$.
**Require:** Dataset $\mathcal{D} = \{\mathbf{x} \in \mathbb{R}^{(n \times d)}, \mathbf{y} \in \mathbb{R}^{(n \times 1)}\}$.
**Require:** Optimization hyperparameters: batch size $b$ and learning rate $\eta$.
**Ensure:** Learned SoftKI coefficients $\alpha$.

$\qquad\qquad\qquad\qquad\qquad\qquad\qquad\qquad\qquad\qquad\qquad\qquad\qquad\quad$ ▷ Model Training

1: $\mathbf{z} \leftarrow \texttt{kmeans}(\mathbf{x}, m)$
2: **for** $i = 1$ **to** epochs **do**
3: $\quad$ **for** $(\mathbf{x}_b, \mathbf{y}_b)$ in $\texttt{batch}(\mathcal{D}, b)$ **do**
4: $\quad\quad$ $\boldsymbol{\Sigma}_{\mathbf{x}_b \mathbf{z}} \leftarrow \texttt{softmax\_interpolation}(\mathbf{x}_b, \mathbf{z})$
5: $\quad\quad$ $\mathbf{K}_{\mathbf{zz}} \leftarrow [\, k(z_i, z_j) \,]_{ij}$
6: $\quad\quad$ $\mathbf{K}_{\mathbf{x}_b \mathbf{x}_b}^{\text{SoftKI}} \leftarrow \boldsymbol{\Sigma}_{\mathbf{x}_b \mathbf{z}} \mathbf{K}_{\mathbf{zz}} \boldsymbol{\Sigma}_{\mathbf{x}_b \mathbf{z}}^{\top}$
7: $\quad\quad$ $\theta \leftarrow \theta + \eta \, \nabla_\theta \log \widehat{p}\Big(\mathbf{y}_b \,\big|\, \mathbf{x}_b;\, \mathbf{K}_{\mathbf{x}_b \mathbf{x}_b}^{\text{SoftKI}} + \boldsymbol{\Lambda}\Big)$ $\qquad\qquad$ ▷ Stabilized MLL (Section 3.2)
8: $\quad$ **end for**
9: **end for**

$\qquad\qquad\qquad\qquad\qquad\qquad\qquad\qquad\qquad\qquad\qquad\qquad$ ▷ Stabilized Inference (Section 3.3

10: $\boldsymbol{\Sigma}_{\mathbf{xz}} \leftarrow \texttt{softmax\_interpolation}(\mathbf{x}, \mathbf{z})$
11: $\mathbf{U}_{\mathbf{zz}}^{\top} \mathbf{U}_{\mathbf{zz}} \leftarrow \texttt{cholesky}(\mathbf{K}_{\mathbf{zz}})$
12: $\mathbf{Q}, \mathbf{R} \leftarrow \texttt{QR}\left(\begin{pmatrix} \boldsymbol{\Lambda}^{-1/2}\, \boldsymbol{\Sigma}_{\mathbf{xz}} \mathbf{K}_{\mathbf{zz}} \\ \mathbf{U}_{\mathbf{zz}} \end{pmatrix}\right)$
13: $\alpha \leftarrow \mathbf{R}^{-1} \mathbf{Q}^{\top} \begin{pmatrix} \boldsymbol{\Lambda}^{-1/2}\, \mathbf{y} \\ 0 \end{pmatrix}$
14: **return** $\alpha$

---

**Stabilized MLL.** We combine SoftKI's MLL with Hutchinson's pseudoloss to arrive at the objective

$$\log \widehat{p}(\mathbf{y} \mid \mathbf{x}; \theta) = \begin{cases} \log p(\mathbf{y} \mid \mathbf{x}; \theta) & \text{when stable} \\ \log \tilde{p}(\mathbf{y} \mid \mathbf{x}; \theta) & \text{otherwise} \end{cases}$$

for optimizing SoftKI's hyperparameters. In practice, this means that we default to using SoftKI's MLL and fallback to Hutchinson's pseudoloss when numeric instability is encountered. This enables SoftKI to accurately recover approximations of the MLL's gradient during gradient-based optimization, even when the current positioning of interpolation points would otherwise make the direct computation of the MLL's gradient unstable, while relying on the exact MLL as much as possible.

**Stochastic optimization.** Instead of computing $\nabla \log \widehat{p}(\mathbf{y} \mid \mathbf{x}; \theta)$ on the entire dataset $\mathbf{x}$, we can compute $\nabla \log \widehat{p}(\mathbf{y} \mid \mathbf{x}_b; \theta)$ on a minibatch of data $\mathbf{x}_b$ of size $b$ to perform stochastic optimization (see line 7 of Algorithm 1). The stochastic estimate provides an unbiased estimator of the gradient. In this way, SoftKI can leverage powerful stochastic optimization techniques used to train neural networks such as Adam (Kingma & Ba, 2014), and hardware acceleration such as graphics processing units (GPUs). The time complexity of evaluating $\nabla \log \widehat{p}(\mathbf{y} \mid \mathbf{x}_b; \theta)$ is cheap, costing $\mathcal{O}(b^2 m)$ operations with space complexity $\mathcal{O}(bm)$. The lower space complexity makes it easier to use GPUs since they have more memory constraints compared to CPUs. We emphasize that we are performing stochastic gradient descent and not stochastic variational inference as in SVGP (Hensman et al., 2013). In particular, we do not define a distribution on the interpolation points $\mathbf{z}$ nor make a variational approximation.

### 3.3 Posterior Inference

Because $\mathbf{K}_{\mathbf{xx}}^{\text{SoftKI}}$ is low-rank, we can use the matrix inversion lemma to rewrite the posterior predictive of SoftKI as

$$p(f(*) \mid \mathbf{x}, \mathbf{y}) = \mathcal{N}(\widehat{\mathbf{K}}_{*\mathbf{z}} \widehat{\mathbf{C}}^{-1} \widehat{\mathbf{K}}_{\mathbf{zx}} \boldsymbol{\Lambda}^{-1} \mathbf{y}, \mathbf{K}_{**}^{\text{SoftKI}} - \mathbf{K}_{*\mathbf{x}}^{\text{SoftKI}} (\boldsymbol{\Lambda}^{-1} - \boldsymbol{\Lambda}^{-1} \widehat{\mathbf{K}}_{\mathbf{xz}} \widehat{\mathbf{C}}^{-1} \widehat{\mathbf{K}}_{\mathbf{zx}} \boldsymbol{\Lambda}^{-1}) \mathbf{K}_{\mathbf{x}*}^{\text{SoftKI}})$$

where $\widehat{\mathbf{C}} = \mathbf{K_{zz}} + \widehat{\mathbf{K}}_{\mathbf{zx}}\boldsymbol{\Lambda}^{-1}\widehat{\mathbf{K}}_{\mathbf{xz}}$ (see Appendix A). To perform posterior mean inference, we solve

$$\widehat{\mathbf{C}}\alpha = \widehat{\mathbf{K}}_{\mathbf{zx}}\boldsymbol{\Lambda}^{-1}\mathbf{y} \tag{9}$$

for weights $\alpha$. Note that this is the posterior mean of a SGPR with $\mathbf{C}$ replaced with $\widehat{\mathbf{C}}$ and $\mathbf{K_{xz}}$ replaced with $\widehat{\mathbf{K}}_{\mathbf{xz}}$. Moreover, note that $\widehat{\mathbf{C}}$ is simply $\mathbf{C}$ with $\mathbf{K_{xz}}$ replaced with $\widehat{\mathbf{K}}_{\mathbf{xz}}$. Since $\widehat{\mathbf{C}}$ is a $m \times m$ matrix, the solution of the system of linear equations has time complexity $\mathcal{O}(m^3)$. The formation of $\widehat{\mathbf{C}}$ requires the multiplication of a $m \times n$ matrix with a $n \times m$ matrix which has time complexity $\mathcal{O}(nm^2)$. Thus the complexity of SoftKI posterior mean inference is $\mathcal{O}(nm^2)$ since it is dominated by the formation of $\widehat{\mathbf{C}}$. We refer the reader to the supplementary material for discussion of the posterior covariance (Appendix A).

**Solving with QR.** Unfortunately, solving Equation 9 for $\alpha$ can be numerically unstable. Foster et al. (2009) introduce a stable QR solver approach for a Subset of Regressors (SoR) GP (Smola & Bartlett, 2000) which we adapt to a SoftKI. Since the form of the SoftKI and SGPR posterior are similar, this QR solver approach can also be adapted to improve the performance of SGPR (see Appendix B).

Define the block matrix

$$\mathbf{A} = \begin{pmatrix} \boldsymbol{\Lambda}^{-1/2}\widehat{\mathbf{K}}_{\mathbf{xz}} \\ \mathbf{U_{zz}} \end{pmatrix}$$

where $\mathbf{U}_{\mathbf{zz}}^{\top}\mathbf{U_{zz}} = \mathbf{K_{zz}}$ is the upper triangular Cholesky decomposition of $\mathbf{K_{zz}}$ so that

$$\mathbf{A}^{\top}\mathbf{A} = \widehat{\mathbf{K}}_{\mathbf{zx}}\boldsymbol{\Lambda}^{-1}\widehat{\mathbf{K}}_{\mathbf{xz}} + \mathbf{K_{zz}} = \widehat{\mathbf{C}}.$$

Let $\mathbf{QR} = \mathbf{A}$ be the QR decomposition of $\mathbf{A}$ so that $\mathbf{Q}$ is a $(n+m) \times m$ orthonormal matrix and $\mathbf{R}$ is $m \times m$ right triangular matrix. Then

$$\widehat{\mathbf{C}}\alpha = \widehat{\mathbf{K}}_{\mathbf{zx}}\boldsymbol{\Lambda}^{-1}\mathbf{y} \tag{$\Longleftrightarrow$}$$

$$(\mathbf{QR})^{\top}(\mathbf{QR})\alpha = (\mathbf{QR})^{\top}\begin{pmatrix}\boldsymbol{\Lambda}^{-1/2}\mathbf{y} \\ 0\end{pmatrix} \tag{$\Longleftrightarrow$}$$

$$\mathbf{R}\alpha = \mathbf{Q}^{\top}\begin{pmatrix}\boldsymbol{\Lambda}^{-1/2}\mathbf{y} \\ 0\end{pmatrix}. \tag{10}$$

We thus solve Equation 10 for $\alpha$ via a triangular solve in $\mathcal{O}(m^2)$ time. For additional experiments demonstrating the inability of other linear solvers to offer comparable accuracy to the QR-stabilized solve detailed in this section, see Appendix B.

## 4 Experiments

We compare the performance of SoftKIs against popular scalable GPs on selected data sets from the `UCI` data set repository (Kelly et al., 2017), a common GP benchmark (Section 4.1). Next, we test SoftKIs on high-dimensional molecule data sets from the domain of computational chemistry (Section 4.2). Finally, we explore the numerical stability of SoftKI (Section 4.3).

### 4.1 Benchmark on UCI Regression

We evaluate the efficacy of a SoftKI against other scalable GP methods on data sets of varying size $n$ and data dimensionality $d$ from the `UCI` repository (Kelly et al., 2017). We choose SGPR and SVGP as two scalable GP methods since these methods can be applied in a relatively blackbox fashion, and thus, can be applied to many data sets. It is not possible to apply SKI to most datasets that we test on due to scalability issues with data dimensionality, and so we omit it. For additional comparisons to other alternative SKI architectures, see Appendix C.3.

**Experiment details.** For this experiment, we split the data set into 0.9 for training and 0.1 for testing. We standardize the data to have mean 0 and standard deviation 1 using the training data set. We use a Matérn 3/2 kernel and a learnable output scale for each dimension (ARD). We choose $m = 512$ inducing

| Dataset | $n$ | $d$ | Exact GP | SoftKI $m = 512$ | SGPR $m = 512$ | SVGP $m = 1024$ |
|---|---|---|---|---|---|---|
| 3droad | 391386 | 3 | $--$ | $0.583 \pm 0.010$ | $--$ | $\mathbf{0.389} \pm 0.001$ |
| Kin40k | 36000 | 8 | $0.039 \pm 0.001$ | $0.169 \pm 0.008$ | $0.177 \pm 0.004$ | $\mathbf{0.165} \pm 0.005$ |
| Protein | 41157 | 9 | $0.044 \pm 0.000$ | $\mathbf{0.596} \pm 0.016$ | $0.602 \pm 0.015$ | $0.607 \pm 0.012$ |
| Houseelectric | 1844352 | 11 | $--$ | $\mathbf{0.047} \pm \mathbf{0.001}$ | $--$ | $\mathbf{0.047} \pm 0.000$ |
| Bike | 15641 | 17 | $0.040 \pm 0.003$ | $\mathbf{0.062} \pm 0.001$ | $0.108 \pm 0.004$ | $0.084 \pm 0.006$ |
| Elevators | 14939 | 18 | $0.108 \pm 0.014$ | $\mathbf{0.360} \pm 0.006$ | $0.395 \pm 0.005$ | $0.384 \pm 0.008$ |
| Keggdirected | 43944 | 20 | $0.070 \pm 0.003$ | $\mathbf{0.080} \pm 0.005$ | $0.099 \pm 0.004$ | $0.082 \pm 0.004$ |
| Pol | 13500 | 26 | $0.035 \pm 0.001$ | $\mathbf{0.075} \pm 0.002$ | $0.127 \pm 0.002$ | $0.122 \pm 0.002$ |
| Keggundirected | 57247 | 27 | $0.110 \pm 0.004$ | $\mathbf{0.115} \pm 0.004$ | $0.128 \pm 0.009$ | $0.121 \pm 0.007$ |
| Buzz | 524925 | 77 | $--$ | $\mathbf{0.240} \pm 0.001$ | $--$ | $0.250 \pm 0.002$ |
| Song | 270000 | 90 | $--$ | $\mathbf{0.777} \pm 0.004$ | $--$ | $0.794 \pm 0.006$ |
| Slice | 48150 | 385 | $--$ | $\mathbf{0.022} \pm 0.006$ | $0.520 \pm 0.001$ | $0.082 \pm 0.001$ |

Table 2: Test RMSE on `UCI` datasets. Best results obtained by an approximate GP are bolded. Exact GP included for reference. Entries marked "$--$" indicate datasets that triggered out of memory errors.

| Dataset | $n$ | $d$ | Exact GP | SoftKI $m = 512$ | SGPR $m = 512$ | SVGP $m = 1024$ |
|---|---|---|---|---|---|---|
| 3droad | 391386 | 3 | $--$ | $0.953 \pm 0.041$ | $--$ | $\mathbf{0.597} \pm 0.008$ |
| Kin40k | 36000 | 8 | $-1.636 \pm 0.006$ | $0.055 \pm 0.043$ | $\mathbf{-0.104} \pm 0.007$ | $-0.082 \pm 0.007$ |
| Protein | 41157 | 9 | $-0.904 \pm 0.008$ | $\mathbf{0.905} \pm 0.026$ | $1.031 \pm 0.009$ | $1.047 \pm 0.009$ |
| Houseelectric | 1844352 | 11 | $--$ | $1.274 \pm 1.425$ | $--$ | $\mathbf{-1.492} \pm 0.007$ |
| Bike | 15641 | 17 | $-1.662 \pm 0.065$ | $-0.379 \pm 0.016$ | $-0.296 \pm 0.011$ | $\mathbf{-0.680} \pm 0.017$ |
| Elevators | 14939 | 18 | $-0.590 \pm 0.075$ | $\mathbf{0.406} \pm 0.021$ | $0.587 \pm 0.006$ | $0.586 \pm 0.009$ |
| Keggdirected | 43944 | 20 | nan | $0.421 \pm 0.063$ | $-0.896 \pm 0.041$ | $\mathbf{-0.934} \pm 0.017$ |
| Pol | 13500 | 26 | $-1.750 \pm 0.020$ | $\mathbf{-0.710} \pm 0.028$ | $-0.428 \pm 0.002$ | $-0.394 \pm 0.012$ |
| Keggundirected | 57247 | 27 | nan | $0.302 \pm 0.136$ | $\mathbf{-0.595} \pm 0.039$ | $-0.590 \pm 0.021$ |
| Buzz | 524925 | 77 | $--$ | $0.276 \pm 0.348$ | $--$ | $\mathbf{0.130} \pm 0.008$ |
| Song | 270000 | 90 | $--$ | $\mathbf{1.179} \pm 0.008$ | $--$ | $1.288 \pm 0.002$ |
| Slice | 48150 | 385 | $--$ | $1.258 \pm 0.149$ | $1.330 \pm 0.002$ | $\mathbf{-0.662} \pm 0.002$ |

Table 3: Test NLL on `UCI` datasets. Best approximate results obtained by an approximate GP are bolded. Exact GP included for reference. Entries marked "$--$" indicate datasets that triggered out of memory errors. Entries marked **nan** indicate that the computations were unstable.

points for SoftKI. Following Wang et al. (2019), we use $m = 512$ for SGPR and $m = 1024$ for SVGP. We learn model hyperparameters for SoftKI by maximizing Hutchinson's pseudoloss, and for SGPR and SVGP by maximizing the ELBO. For reference, we also include results for a GP fit with block CG, termed an Exact GP. For Exact GP we use the `KeOps` (Charlier et al., 2021) library integration with `GPyTorch` to handle the additional scaling challenges of working with the full $n \times n$ kernel.

We perform 50 epochs of training using the Adam optimizer (Kingma & Ba, 2014) for all methods with a learning rate of $\eta = 0.01$. The learning rate for SGPR is $\eta = 0.1$ since we are not performing batching. We use a default implementation of SGPR and SVGP from `GPyTorch`. For SoftKI and SVGP, we use a minibatch size of 1024. We report the average test RMSE (Table 2) and NLL (Table 3) for each method across three seeds. Additional timing information is given in Appendix C.2.

**Results.** We observe that SoftKI has competitive test RMSE performance compared to SGPR and SVGP, exceeding them when the dimension is modest ($d \approx 10$). For the test NLL, we observe cases where SoftKI's test NLL is smaller or larger relative to the test RMSE. As a reminder, the NLL for a GP method consists of two components: the RMSE and the complexity of the model. Consequently, a small test NLL relative to the test RMSE indicates a relatively small amount of noise in the data set. Conversely, a large test NLL

| Dataset | $n$ | $d$ | Exact GP | SoftKI $m$=512 | SGPR $m$=512 | SVGP $m$=1024 |
|---|---|---|---|---|---|---|
| Ac-ala3-nhme | 76598 | 126 | $0.017 \pm 0.000$ | $\mathbf{0.790} \pm 0.010$ | $0.852 \pm 0.005$ | $0.836 \pm 0.005$ |
| Dha | 62777 | 168 | $0.016 \pm 0.000$ | $0.886 \pm 0.012$ | $0.897 \pm 0.012$ | $\mathbf{0.882} \pm 0.010$ |
| Stachyose | 24544 | 261 | $0.020 \pm 0.000$ | $\mathbf{0.363} \pm 0.012$ | $0.643 \pm 0.002$ | $0.555 \pm 0.001$ |
| At-at | 18000 | 354 | $0.020 \pm 0.001$ | $\mathbf{0.528} \pm 0.011$ | $0.591 \pm 0.011$ | $0.558 \pm 0.009$ |
| At-at-cg-cg | 9137 | 354 | $0.021 \pm 0.000$ | $0.502 \pm 0.007$ | $0.565 \pm 0.029$ | $\mathbf{0.461} \pm 0.026$ |
| Buckyball-catcher | 5491 | 444 | $0.018 \pm 0.001$ | $\mathbf{0.153} \pm 0.006$ | $0.393 \pm 0.008$ | $0.307 \pm 0.015$ |
| DW-nanotube | 4528 | 1110 | $0.012 \pm 0.000$ | $\mathbf{0.031} \pm 0.001$ | $1.001 \pm 0.048$ | $0.045 \pm 0.000$ |

Table 4: Test RMSE on `MD22` datasets. Best results obtained by an approximate GP are bolded. Exact GP results are included for reference. Exact GPs are roughly $20-875\times$ slower than SoftKI (see Table 10), and thus, only run for 50 epochs.

| Dataset | $n$ | $d$ | Exact GP | SoftKI $m$=512 | SGPR $m$=512 | SVGP $m$=1024 |
|---|---|---|---|---|---|---|
| Ac-ala3-nhme | 76598 | 126 | $0.000 \pm 0.000$ | $\mathbf{1.188} \pm 0.014$ | $1.367 \pm 0.003$ | $1.356 \pm 0.002$ |
| Dha | 62777 | 168 | $-0.975 \pm 0.844$ | $\mathbf{1.299} \pm 0.014$ | $1.418 \pm 0.006$ | $1.409 \pm 0.007$ |
| Stachyose | 24544 | 261 | $-1.330 \pm 0.003$ | $\mathbf{0.522} \pm 0.059$ | $1.123 \pm 0.002$ | $1.012 \pm 0.003$ |
| At-at | 18000 | 354 | $-1.418 \pm 0.002$ | $\mathbf{0.795} \pm 0.007$ | $1.041 \pm 0.006$ | $1.003 \pm 0.008$ |
| At-at-cg-cg | 9137 | 354 | $-1.426 \pm 0.000$ | $\mathbf{0.742} \pm 0.010$ | $0.990 \pm 0.024$ | $0.800 \pm 0.016$ |
| Buckyball-catcher | 5491 | 444 | $-1.567 \pm 0.003$ | $\mathbf{-0.278} \pm 0.061$ | $0.689 \pm 0.005$ | $0.449 \pm 0.015$ |
| DW-nanotube | 4528 | 1110 | $-1.992 \pm 0.000$ | $\mathbf{-1.551} \pm 0.049$ | $1.520 \pm 0.021$ | $-0.954 \pm 0.009$ |

Table 5: Test NLL on `MD22` datasets. Best results obtained by an approximate GP are bolded. Exact GP results are included for reference. Exact GPs are roughly $20-875\times$ slower than SoftKI (see Table 10), and thus, only run for 50 epochs.

relative to the test RMSE indicates a large amount of noise in the dataset. We can see instances of this in the `bike` dataset where SoftKI achieves comparable test RMSE to SGPR and SVGP, but results in larger NLL. This indicates that SoftKI's kernel is more complex compared to the variational GP kernels.

## 4.2 High-Dimensional Molecule Dataset

Since SoftKI's are performant on high dimensional data sets from the `UCI` repository, we also test the performance on molecular potential energy surface data on the `MD22` (Chmiela et al., 2023) dataset. These are high-dimensional datasets that give the geometric coordinates of atomic nuclei in biomolecules and their respective energies.

**Experiment details.** For consistency, we keep the experimental setup the same as for `UCI` regression but up the number of epochs of training to 200 due to slower convergence on these datasets. We still use the Matérn kernel and not a molecule-specific kernel that incorporates additional information such as the atomic number of each atom (*e.g.*, a Hydrogen atom) or invariances (*e.g.*, rotational invariance). We standardize the data set to have mean 0 and standard deviation 1. We note that in an actual application of GP regression to this setting, we may only opt to center the targets to be mean 0. This is because energy is a relative number that can be arbitrarily shifted, whereas scaling the distances between atoms will affect the physics. As before, we include results for Exact GPs as a reference. Since they are roughly 20 to 875 times slower than SoftKI, so we only run them for 50 epochs (see Table 10).

**Results.** Table 4 and Table 5 compares the test RMSE and test NLL of various GP models trained on `MD22`. We see that SoftKI is a competitive approximate method on these datasets, especially on the test NLL. GPs that fit forces have been successfully applied to fit such data sets to chemical accuracy. In our case, we do not fit derivative information.

| Dataset | Test RMSE | | Test NLL | |
|---|---|---|---|---|
| | MLL | Hutchinson Pseduoloss | MLL | Hutchinson Pseduoloss |
| 3droad | nan | **0.541 ± 0.005** | nan | **1.004 ± 0.175** |
| Kin40k | **0.169 ± 0.008** | 0.183 ± 0.006 | **0.055 ± 0.043** | 0.07 ± 0.029 |
| Protein | **0.596 ± 0.016** | 0.602 ± 0.016 | **0.904 ± 0.03** | 0.914 ± 0.028 |
| Houseelectric | nan | **0.049 ± 0.002** | nan | **1.275 ± 1.103** |
| Bike | nan | **0.073 ± 0.005** | nan | **-0.101 ± 0.179** |
| Elevators | **0.36 ± 0.007** | **0.36 ± 0.007** | **0.404 ± 0.024** | 0.404 ± 0.025 |
| Keggdirected | **0.08 ± 0.005** | 0.082 ± 0.004 | **0.293 ± 0.169** | 0.628 ± 0.289 |
| Pol | **0.075 ± 0.002** | 0.084 ± 0.001 | -0.694 ± 0.06 | **-0.783 ± 0.033** |
| Keggundirected | **0.114 ± 0.005** | 0.118 ± 0.004 | 0.308 ± 0.025 | **0.291 ± 0.089** |
| Buzz | nan | **0.24 ± 0.0** | nan | **0.599 ± 0.555** |
| Song | **0.777 ± 0.004** | **0.777 ± 0.004** | 1.179 ± 0.008 | **1.178 ± 0.007** |
| Slice | **0.022 ± 0.006** | 0.044 ± 0.005 | 1.328 ± 0.045 | **0.607 ± 0.058** |

Table 6: Comparison of SoftKI test RMSE and NLL when training with the SoftKI MLL $\log p(\mathbf{y} \mid \mathbf{x}; \boldsymbol{\theta})$ versus the Hutchinson pseudoloss $\log \tilde{p}(\mathbf{y} \mid \mathbf{x}; \boldsymbol{\theta})$. Best results are bolded. Entries labeled nan, represent instances where using the exact MLL resulted in numerical instability in SoftKI (see Section 4.3).

### 4.3 Numerical Stability

In Section 3.2, we advocated for the use of Hutchinson's pseudoloss to overcome numerical stability issues that arise when calculating the SoftKI MLL. This adjustment is specifically to address situations where the matrix $\mathbf{K}_{\mathbf{x}_b \mathbf{x}_b}^{\text{SoftKI}} + \boldsymbol{\Lambda}$ is not positive semi-definite in float32 precision. In other approximate GP models implemented in GPyTorch, this challenge is typically addressed by performing a Cholesky decomposition followed by an efficient low-rank computation of the log determinant through the matrix determinant lemma.

In our experience, even when using versions of the Cholesky decomposition that add additional jitter along the diagonal there are still situations where $\mathbf{K}_{\mathbf{zz}}$ can be poorly conditioned, particularly when $n$ is large. We conjecture that this behavior originates from situations where the learned interpolation points of a SoftKI coincide at similar positions, driving the effective rank of $\mathbf{K}_{\mathbf{zz}}$ down. In these situations, Hutchinson's pseudoloss offers a more stable alternative because it does not directly rely on the matrix being invertible.

In Table 6, we replicate the UCI experiment of Section 4.1 using the SoftKI MLL $\log p(\mathbf{y} \mid \mathbf{x}; \theta)$ and Hutchinson's pseudoloss $\log \tilde{p}(\mathbf{y} \mid \mathbf{x}; \theta)$. For most datasets, optimizing with the SoftKI MLL produces better test RMSE. However, more numericaly instability is also encountered, with SoftKI failing on four datasets that we tested. In these cases, we find that Cholesky decomposition fails.

## 5 Conclusion

In this paper, we introduce SoftKI, an approximate GP designed for regression on large and high-dimensional datasets. SoftKI combines aspects of SKI and inducing points methods to retain the benefits of kernel interpolation while also scaling to higher dimensional datasets. We have tested SoftKI on a variety of datasets and shown that it is possible to perform kernel interpolation in high dimensional spaces in a way that is competitive with other approximate GP abstractions that leverage inducing points. Exploring methods that enforce stricter sparsity in the interpolation matrices, such as through thresholding, could enable the use of sparse matrix algebra further improving the cost of inference.

**Acknowledgements** Chris Camaño acknowledges support by the National Science Foundation Graduate Research Fellowship under Grant No. 2139433, and the Kortschak scholars program. Additionally, we would like to extend our appreciation to Ethan N. Epperly for his thoughtful comments and advice on this work.

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

## A  SoftKI Posterior Predictive Derivation

As a reminder, the SoftKI posterior predictive is

$$p(f(*) \,|\, \mathbf{x}, \mathbf{y}) = \mathcal{N}(\widehat{\mathbf{K}}_{*\mathbf{z}}\widehat{\mathbf{C}}^{-1}\widehat{\mathbf{K}}_{\mathbf{zx}}\mathbf{\Lambda}^{-1}\mathbf{y}, \mathbf{K}_{**}^{\text{SoftKI}} - \mathbf{K}_{*\mathbf{x}}^{\text{SoftKI}}(\mathbf{\Lambda}^{-1} - \mathbf{\Lambda}^{-1}\widehat{\mathbf{K}}_{\mathbf{xz}}\widehat{\mathbf{C}}^{-1}\widehat{\mathbf{K}}_{\mathbf{zx}}\mathbf{\Lambda}^{-1})\mathbf{K}_{\mathbf{x}*}^{\text{SoftKI}}) \tag{11}$$

where $\widehat{\mathbf{C}} = \mathbf{K}_{\mathbf{zz}} + \mathbf{K}_{\mathbf{zx}}\mathbf{\Lambda}^{-1}\mathbf{K}_{\mathbf{xz}}$. To derive this, we first recall how to apply the matrix inversion lemma to the posterior mean of a SGPR, which as a reminder uses the covariance approximation $\mathbf{K}_{\mathbf{xx}} \approx \mathbf{K}_{\mathbf{xx}}^{\text{SGPR}} = \mathbf{K}_{\mathbf{xz}}\mathbf{K}_{\mathbf{zz}}^{-1}\mathbf{K}_{\mathbf{zx}}$

**Lemma A.1** (Matrix inversion with GPs).

$$\mathbf{K}_{*\mathbf{z}}(\mathbf{K}_{\mathbf{zz}} + \mathbf{K}_{\mathbf{zx}}\mathbf{\Lambda}^{-1}\mathbf{K}_{\mathbf{xz}})^{-1}\mathbf{K}_{\mathbf{zx}}\mathbf{\Lambda}^{-1} = \mathbf{K}_{*\mathbf{x}}^{SGPR}(\mathbf{K}_{\mathbf{xx}}^{SGPR} + \mathbf{\Lambda})^{-1} \tag{12}$$

*Proof.*

$$
\begin{aligned}
&\mathbf{K}_{\mathbf{xx}}^{\text{SGPR}} + \mathbf{\Lambda} = \mathbf{K}_{\mathbf{xx}}^{\text{SGPR}} + \mathbf{\Lambda} && \text{(identity)}\\
\iff &\mathbf{I} = (\mathbf{K}_{\mathbf{xx}}^{\text{SGPR}} + \mathbf{\Lambda})^{-1}\mathbf{K}_{\mathbf{xx}}^{\text{SGPR}} + (\mathbf{K}_{\mathbf{xx}}^{\text{SGPR}} + \mathbf{\Lambda})^{-1}\mathbf{\Lambda} && \text{(mult by } (\mathbf{K}_{\mathbf{xx}}^{\text{SGPR}} + \mathbf{\Lambda})^{-1})\\
\iff &\mathbf{I} - (\mathbf{K}_{\mathbf{xx}}^{\text{SGPR}} + \mathbf{\Lambda})^{-1}\mathbf{K}_{\mathbf{xx}}^{\text{SGPR}} = (\mathbf{K}_{\mathbf{xx}}^{\text{SGPR}} + \mathbf{\Lambda})^{-1}\mathbf{\Lambda} && \text{(rearrange)}
\end{aligned}
$$

$$\iff \mathbf{K}_{*\mathbf{x}}^{\text{SGPR}} - \mathbf{K}_{*\mathbf{x}}^{\text{SGPR}}(\mathbf{K}_{\mathbf{xx}}^{\text{SGPR}} + \mathbf{\Lambda})^{-1}\mathbf{K}_{\mathbf{xx}}^{\text{SGPR}} = \mathbf{K}_{*\mathbf{x}}^{\text{SGPR}}(\mathbf{K}_{\mathbf{xx}}^{\text{SGPR}} + \mathbf{\Lambda})^{-1}\mathbf{\Lambda}$$
$$\text{(mult both sides by } \mathbf{K}_{*\mathbf{x}}^{\text{SGPR}} \text{ on left)}$$

$$\iff \mathbf{K}_{*\mathbf{z}}\mathbf{K}_{\mathbf{zz}}^{-1}\mathbf{K}_{\mathbf{zx}} - \mathbf{K}_{*\mathbf{z}}\mathbf{K}_{\mathbf{zz}}^{-1}\mathbf{K}_{\mathbf{zx}}(\mathbf{K}_{\mathbf{xz}}\mathbf{K}_{\mathbf{zz}}^{-1}\mathbf{K}_{\mathbf{zx}} + \mathbf{\Lambda})^{-1}\mathbf{K}_{\mathbf{xz}}\mathbf{K}_{\mathbf{zz}}^{-1}\mathbf{K}_{\mathbf{zx}} = \mathbf{K}_{*\mathbf{x}}^{\text{SGPR}}(\mathbf{K}_{\mathbf{xx}}^{\text{SGPR}} + \mathbf{\Lambda})^{-1}\mathbf{\Lambda}$$
$$\text{(defn } \mathbf{K}_{\mathbf{xx}}^{\text{SGPR}})$$

$$\iff \mathbf{K}_{*\mathbf{z}}(\mathbf{K}_{\mathbf{zz}}^{-1} - \mathbf{K}_{\mathbf{zz}}^{-1}\mathbf{K}_{\mathbf{zx}}(\mathbf{K}_{\mathbf{xz}}\mathbf{K}_{\mathbf{zz}}^{-1}\mathbf{K}_{\mathbf{zx}} + \mathbf{\Lambda})^{-1}\mathbf{K}_{\mathbf{xz}}\mathbf{K}_{\mathbf{zz}}^{-1})\mathbf{K}_{\mathbf{zx}} = \mathbf{K}_{*\mathbf{x}}^{\text{SGPR}}(\mathbf{K}_{\mathbf{xx}}^{\text{SGPR}} + \mathbf{\Lambda})^{-1}\mathbf{\Lambda} \quad \text{(factor)}$$

$$\iff \mathbf{K}_{*\mathbf{z}}(\mathbf{K}_{\mathbf{zz}} + \mathbf{K}_{\mathbf{zx}}\mathbf{\Lambda}^{-1}\mathbf{K}_{\mathbf{xz}})^{-1}\mathbf{K}_{\mathbf{zx}} = \mathbf{K}_{*\mathbf{x}}^{\text{SGPR}}(\mathbf{K}_{\mathbf{xx}}^{\text{SGPR}} + \mathbf{\Lambda})^{-1}\mathbf{\Lambda} \quad \text{(matrix inversion lemma)}$$

$$\iff \mathbf{K}_{*\mathbf{z}}(\mathbf{K}_{\mathbf{zz}} + \mathbf{K}_{\mathbf{zx}}\mathbf{\Lambda}^{-1}\mathbf{K}_{\mathbf{xz}})^{-1}\mathbf{K}_{\mathbf{zx}}\mathbf{\Lambda}^{-1} = \mathbf{K}_{*\mathbf{x}}^{\text{SGPR}}(\mathbf{K}_{\mathbf{xx}}^{\text{SGPR}} + \mathbf{\Lambda})^{-1}. \quad \text{(mult } \mathbf{\Lambda}^{-1} \text{ on right)}$$

$$\square$$

Now, we extend the matrix inversion lemma to SoftKI.

**Lemma A.2** (Matrix inversion with interpolation).

$$\widehat{\mathbf{K}}_{*\mathbf{z}}(\mathbf{K}_{\mathbf{zz}} + \widehat{\mathbf{K}}_{\mathbf{zx}}\mathbf{\Lambda}^{-1}\widehat{\mathbf{K}}_{\mathbf{xz}})^{-1}\widehat{\mathbf{K}}_{\mathbf{zx}}\mathbf{\Lambda}^{-1} = \mathbf{K}_{*\mathbf{x}}^{SoftKI}(\mathbf{K}_{\mathbf{xx}}^{SoftKI} + \mathbf{\Lambda})^{-1} \tag{13}$$

*Proof.* Recall $\mathbf{K}_{**}^{\text{SoftKI}} = \mathbf{\Sigma}_{\mathbf{xz}}\mathbf{K}_{\mathbf{zz}}\mathbf{\Sigma}_{\mathbf{zx}} = \mathbf{\Sigma}_{\mathbf{xz}}\mathbf{K}_{\mathbf{zz}}\mathbf{K}_{\mathbf{zz}}^{-1}\mathbf{K}_{\mathbf{zz}}\mathbf{\Sigma}_{\mathbf{zx}} = \widehat{\mathbf{K}}_{\mathbf{xz}}\mathbf{K}_{\mathbf{zz}}^{-1}\widehat{\mathbf{K}}_{\mathbf{zx}}$. The result follows by applying Lemma A.1 with $\mathbf{K}_{\mathbf{xz}}$ replaced with $\widehat{\mathbf{K}}_{\mathbf{xz}}$. $\square$

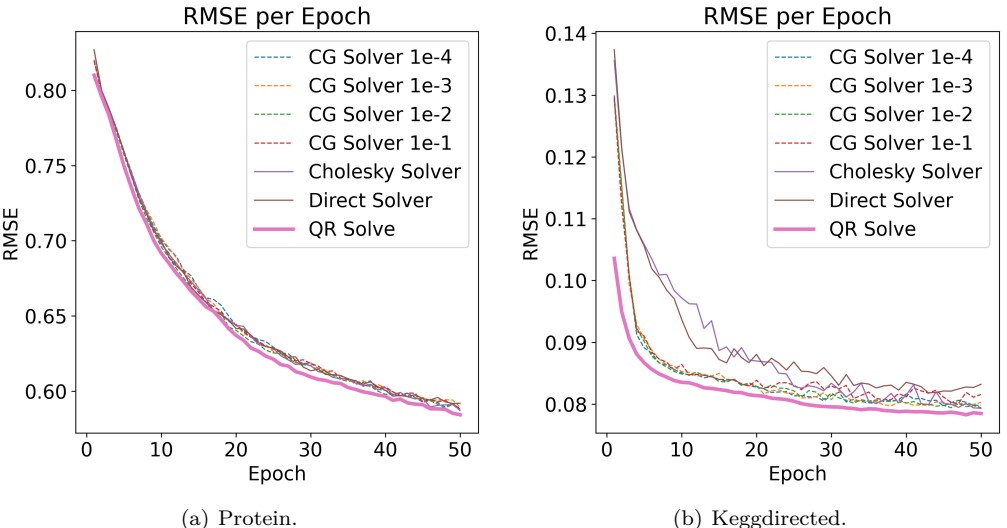

Figure 3: Test RMSE of SoftKI models trained using different linear solvers described in Section B on the `keggundirected` dataset (*left*) and the `protein` dataset (*right*).

The posterior covariance is computed by a simple application of the matrix inversion lemma. To compute it, observe that the same procedure for solving the posterior mean can also be used for solving the posterior covariance by replacing $\mathbf{y}$ with $\mathbf{K}_{\mathbf{x}*}^{\mathrm{SoftKI}}$. More concretely,

$$\mathbf{K}_{**}^{\mathrm{SoftKI}} - \mathbf{K}_{*\mathbf{x}}^{\mathrm{SoftKI}}(\mathbf{\Lambda}^{-1} - \mathbf{\Lambda}^{-1}\widehat{\mathbf{K}}_{\mathbf{xz}}\widehat{\mathbf{C}}^{-1}\widehat{\mathbf{K}}_{\mathbf{zx}}\mathbf{\Lambda}^{-1})\mathbf{K}_{\mathbf{x}*}^{\mathrm{SoftKI}} \tag{14}$$

$$= \mathbf{K}_{**}^{\mathrm{SoftKI}} - \mathbf{K}_{*\mathbf{x}}^{\mathrm{SoftKI}}\mathbf{\Lambda}^{-1}\mathbf{K}_{\mathbf{x}*}^{\mathrm{SoftKI}} + \mathbf{K}_{*\mathbf{x}}^{\mathrm{SoftKI}}\mathbf{\Lambda}^{-1}\widehat{\mathbf{K}}_{\mathbf{xz}}\widehat{\mathbf{C}}^{-1}\widehat{\mathbf{K}}_{\mathbf{zx}}\mathbf{\Lambda}^{-1}\mathbf{K}_{\mathbf{x}*}^{\mathrm{SoftKI}} \tag{15}$$

$$= \mathbf{K}_{**}^{\mathrm{SoftKI}} - \mathbf{K}_{*\mathbf{x}}^{\mathrm{SoftKI}}\mathbf{\Lambda}^{-1}\mathbf{K}_{\mathbf{x}*}^{\mathrm{SoftKI}} + \mathbf{K}_{*\mathbf{x}}^{\mathrm{SoftKI}}\mathbf{\Lambda}^{-1}\widehat{\mathbf{K}}_{\mathbf{xz}}\alpha_* \tag{16}$$

where

$$\widehat{\mathbf{C}}\alpha_* = \widehat{\mathbf{K}}_{\mathbf{zx}}\mathbf{\Lambda}^{-1}\mathbf{K}_{\mathbf{x}*}^{\mathrm{SoftKI}} \tag{17}$$

can be solved for $\alpha_*$ in the same way we solved for the posterior mean (with $\mathbf{y}$ instead of $\mathbf{K}_{\mathbf{x}*}^{\mathrm{SoftKI}}$). Since this depends on the inference point $*$, we cannot precompute the result ahead of time as we could with the posterior mean. Nevertheless, the intermediate results of the QR decomposition can be computed once during posterior mean inference and reused for the covariance prediction. Thus, the time complexity of posterior covariance inference is $\mathcal{O}(nm^2)$ per a test point.

## B  Alternative Methods for Posterior Inference

Section 3.3 details the adaptation of the QR stabilized linear solve for approximate kernels (Foster et al., 2009) to the SoftKI setting. In this section, we provide empirical evidence that illustrates how other linear solvers fail when confronted with datasets that generate noisy kernels. We focus on two datasets, `protein` and `keggundirected`.

As a reminder the QR stabilized linear solve is motivated by the challenge of solving the linear system:

$$\hat{\mathbf{C}}\,\boldsymbol{\alpha} = \hat{\mathbf{K}}_{\mathbf{zx}}\,\mathbf{\Lambda}^{-1}\,\mathbf{y}, \tag{18}$$

where $\hat{\mathbf{C}}$ is the estimated (potentially noisy) kernel matrix $\hat{\mathbf{C}} = \mathbf{K}_{\mathbf{zz}} + \hat{\mathbf{K}}_{\mathbf{zx}}\mathbf{\Lambda}^{-1}\hat{\mathbf{K}}_{\mathbf{xz}}$, and $\hat{\mathbf{K}}_{\mathbf{xz}} = \mathbf{\Sigma}_{\mathbf{xz}}\mathbf{K}_{\mathbf{zz}}$ is the interpolated kernel between interpolations points $\mathbf{z}$ and training points $\mathbf{x}$.

| Dataset | $n$ | $d$ | SoftKI $m$=512 | SGPR $m$=512 | SGPR $m$=512 QR |
|---|---|---|---|---|---|
| Pol | 13500 | 26 | $0.075 \pm 0.002$ | $0.127 \pm 0.002$ | $\mathbf{0.121 \pm 0.002}$ |
| Elevators | 14939 | 18 | $0.36 \pm 0.006$ | $0.395 \pm 0.005$ | $\mathbf{0.392 \pm 0.005}$ |
| Bike | 15641 | 17 | $0.062 \pm 0.001$ | $0.108 \pm 0.004$ | $\mathbf{0.105 \pm 0.004}$ |
| Kin40k | 36000 | 8 | $0.169 \pm 0.008$ | $0.177 \pm 0.004$ | $\mathbf{0.169 \pm 0.004}$ |
| Protein | 41157 | 9 | $0.596 \pm 0.016$ | $0.602 \pm 0.015$ | $\mathbf{0.601 \pm 0.015}$ |
| Keggdirected | 43944 | 20 | $0.08 \pm 0.005$ | $0.099 \pm 0.004$ | $\mathbf{0.082 \pm 0.006}$ |
| Slice | 48150 | 385 | $0.022 \pm 0.006$ | $0.52 \pm 0.001$ | $\mathbf{0.513 \pm 0.001}$ |
| Keggundirected | 57247 | 27 | $0.115 \pm 0.004$ | $0.128 \pm 0.009$ | $\mathbf{0.124 \pm 0.007}$ |
| 3droad | 391386 | 3 | $0.583 \pm 0.01$ | $--$ | $--$ |
| Song | 270000 | 90 | $0.777 \pm 0.004$ | $--$ | $--$ |
| Buzz | 524925 | 77 | $0.24 \pm 0.001$ | $--$ | $--$ |
| Houseelectric | 1844352 | 11 | $0.047 \pm 0.001$ | $--$ | $--$ |

Table 7: Test RMSE on the `UCI` dataset using vanilla SGPR with SGPR using the QR stabilized linear solve described in section 3.3. Reference results for SoftKI are given in gray.

| Method | Learning rate ($\eta$) | Kernel | Starting noise ($\beta$) | Inducing points ($m$) | Test RMSE |
|---|---|---|---|---|---|
| SGPR | 0.1 | Matern $\nu = 1.5$ | 0 | 128 K-means centroids | $6 \times 10^{-3}$ |
| SKI | 0.1 | Matern $\nu = 1.5$ | 0.1 | $20 \times 20$ grid | $6 \times 10^{-3}$ |
| SoftKI | 0.5 | Matern $\nu = 1.5$ | 0.5 | 128 K-means centroids | $2 \times 10^{-3}$ |

Table 8: Experimental settings for the Ricker Wavlet experiment.

In this example we evaluate a direct solver, the Cholesky decomposition method, and CG with convergence tolerances set to $1 \times 10^{-1}$, $1 \times 10^{-2}$, $1 \times 10^{-3}$, and $1 \times 10^{-4}$. Despite adjusting the tolerances for the CG method, our experiments revealed that all solvers—except for the QR-stabilized routine—resulted in training instability. Figure 3 depicts the test RMSE performance across the different solvers. The direct and Cholesky solvers, while more stable than the CG solvers, failed to produce reliable solutions. Similarly, the CG method did not converge to acceptable solutions within the tested tolerance levels. In contrast, the QR-stabilized solver consistently produced stable and accurate solutions justifying its choice as a correctional measure that stabilizes a SoftKI on difficult problems.

Since the shape of the SoftKI posterior is similar to the SGPR posterior (see Lemma A.2), we can also adapt the QR procedure used for SoftKI to SGPR. Table 7 illustrates the results of using QR solving for SGPR. We see that QR solving improves the test RMSE for SGPR. However, unlike SoftKI where QR solving is more important for training stability, we did not observe instability with an inversion of $\mathbf{K}_{\mathbf{xx}}^{\mathrm{SGPR}} + \mathbf{\Lambda}$ using the matrix inversion lemma.

# C  Additional Experiments

We provide supplemental data for experiments in the main text (Section C.1 and Section C.2). We also report comparisons with SKI-based methods (Section C.3) and additional experimental results (Section C.4).

## C.1  Ricker Wavelet Experiment

In Section 3.2 we present an illustrative comparison of SGPR, SKI and SoftKI on the task of learning the Ricker wavelet. Figure 4 visualizes the absolute error given as surfaces. Each model was trained for 100 epochs using the `Adam` optimizer together with a `StepLR` learning-rate scheduler. The dataset is comprised of 3,000 training points and 200 test points. We obtain the initial inducing and interpolation locations for both SGPR and SoftKI by clustering the training inputs with `KMeans`.

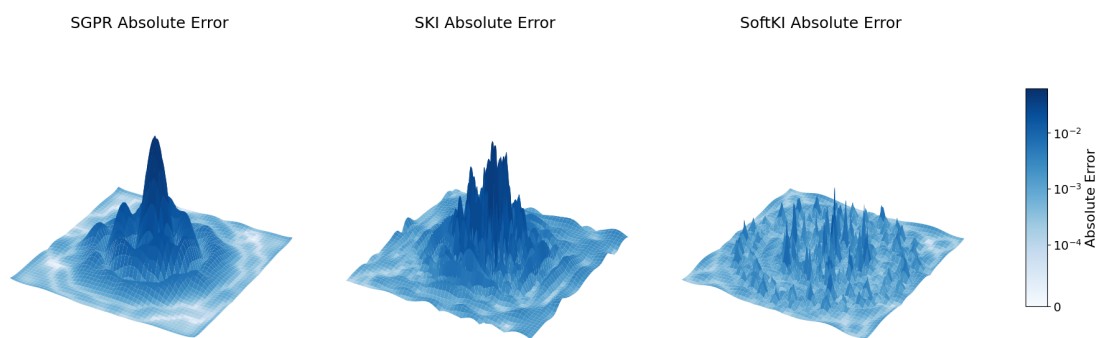

Figure 4: **(Absolute Error Surfaces):** Comparison of the absolute error on the training set domain from Figure 2 visualized as surfaces.

| Dataset | $n$ | $d$ | Exact GP | SoftKI $m = 512$ | SGPR $m = 512$ | SVGP $m = 1024$ |
|---|---|---|---|---|---|---|
| 3droad | 391386 | 3 | —— | $\mathbf{5.586 \pm 0.120}$ | —— | $8.529 \pm 0.216$ |
| Kin40k | 36000 | 8 | $0.720 \pm 0.057$ | $0.453 \pm 0.012$ | $\mathbf{0.019 \pm 0.000}$ | $0.823 \pm 0.010$ |
| Protein | 41157 | 9 | $0.612 \pm 0.015$ | $0.520 \pm 0.006$ | $\mathbf{0.020 \pm 0.000}$ | $0.904 \pm 0.020$ |
| Houseelectric | 1844352 | 11 | —— | $\mathbf{26.066 \pm 1.335}$ | —— | $40.330 \pm 1.182$ |
| Bike | 15641 | 17 | $0.112 \pm 0.001$ | $0.227 \pm 0.011$ | $\mathbf{0.013 \pm 0.000}$ | $0.367 \pm 0.006$ |
| Elevators | 14939 | 18 | $0.261 \pm 0.012$ | $0.178 \pm 0.004$ | $\mathbf{0.013 \pm 0.000}$ | $0.352 \pm 0.046$ |
| Keggdirected | 43944 | 20 | $3.904 \pm 0.208$ | $0.554 \pm 0.004$ | $\mathbf{0.021 \pm 0.000}$ | $0.950 \pm 0.011$ |
| Pol | 13500 | 26 | $0.340 \pm 0.023$ | $0.171 \pm 0.010$ | $\mathbf{0.012 \pm 0.001}$ | $0.325 \pm 0.017$ |
| Keggundirected | 57247 | 27 | $4.337 \pm 0.402$ | $0.722 \pm 0.043$ | $\mathbf{0.026 \pm 0.000}$ | $1.217 \pm 0.047$ |
| Buzz | 524925 | 77 | —— | $\mathbf{7.107 \pm 0.053}$ | —— | $10.974 \pm 0.131$ |
| Song | 270000 | 90 | —— | $\mathbf{3.047 \pm 0.055}$ | —— | $5.851 \pm 0.185$ |
| Slice | 48150 | 385 | —— | $1.363 \pm 0.004$ | $\mathbf{0.029 \pm 0.000}$ | $1.069 \pm 0.014$ |

Table 9: Timing per epoch of hyperparameter optimization in seconds.

## C.2 Supplemental to Experiments

**Hardware details.** We run all experiments on a single Nvidia RTX 3090 GPU which has 24Gb of VRAM. A GPU with more VRAM can support larger datasets. Our machine uses an Intel i9-10900X CPU at 3.70GHz with 10 cores. This primarily affects the timing of SoftKI and SVGP which use batched hyperparameter optimization, and thus, move data on and off the GPU more frequently than SGPR.

**Timing.** Table 9 compares the average training time per epoch in seconds to for hyperparameter optimization of SoftKI vs SVGP. Note that SGPR is much faster but does not support batched stochastic optimization, and thus, is limited to smaller datasets. We observe that SVGP and SoftKI have similar performance characteristics due to the mini-batch gradient descent approach that both SVGP and SoftKI employ. SVGP requires slightly more compute since it additionally learns the parameters of a variational Gaussian distribution.

## C.3 Comparison with SKI-based Methods

As we mentioned in the main text, it is not possible to apply SKI to a majority of the datasets in our experiments due to dimensionality scaling issues. Nevertheless, there are some datasets with smaller dimensionality that SKI-variants such as SKIP (Gardner et al., 2018b) and Simplex-SKI (Kapoor et al., 2021) can be applied to.

| Dataset | $n$ | $d$ | Exact GP | SoftKI $m$=512 | SGPR $m$=512 | SVGP $m$=1024 |
|---|---|---|---|---|---|---|
| Ac-ala3-nhme | 76598 | 126 | $21.300 \pm 0.502$ | $1.113 \pm 0.095$ | $\mathbf{0.035 \pm 0.000}$ | $3.257 \pm 0.406$ |
| Dha | 62777 | 168 | $36.841 \pm 0.375$ | $1.195 \pm 0.017$ | $\mathbf{0.031 \pm 0.000}$ | $2.364 \pm 0.078$ |
| Stachyose | 24544 | 261 | $15.747 \pm 0.172$ | $0.534 \pm 0.011$ | $\mathbf{0.017 \pm 0.001}$ | $0.924 \pm 0.029$ |
| At-at | 18000 | 354 | $2.669 \pm 0.036$ | $0.365 \pm 0.012$ | $\mathbf{0.014 \pm 0.000}$ | $0.699 \pm 0.008$ |
| At-at-cg-cg | 9137 | 354 | $97.700 \pm 4.388$ | $0.295 \pm 0.027$ | $\mathbf{0.013 \pm 0.000}$ | $0.368 \pm 0.023$ |
| Buckyball-catcher | 5491 | 444 | $40.875 \pm 3.186$ | $0.175 \pm 0.005$ | $\mathbf{0.011 \pm 0.002}$ | $0.221 \pm 0.007$ |
| DW-nanotube | 4528 | 1110 | $298.096 \pm 1.270$ | $0.344 \pm 0.026$ | $\mathbf{0.012 \pm 0.002}$ | $0.198 \pm 0.002$ |

Table 10: Timing per epoch of hyperparameter optimization in seconds on MD22.

| | Dataset | n | d | Exact GP | SoftKI $m$=512 | SKIP | Simplex-SKI |
|---|---|---|---|---|---|---|---|
| RMSE | Protein | 41157 | 9 | $0.511 \pm 0.009$ | $0.652 \pm 0.012$ | $0.817 \pm 0.012$ | $\mathbf{0.571 \pm 0.003}$ |
| | Houseelectric | 1844352 | 11 | $0.054 \pm 0.000$ | $\mathbf{0.052 \pm 0.0}$ | $--$ | $0.079 \pm 0.002$ |
| | Elevators | 14939 | 18 | $0.399 \pm 0.011$ | $\mathbf{0.423 \pm 0.011}$ | $0.447 \pm 0.037$ | $0.510 \pm 0.018$ |
| | Keggdirected | 43944 | 20 | $0.083 \pm 0.001$ | $\mathbf{0.089 \pm 0.001}$ | $0.487 \pm 0.005$ | $0.095 \pm 0.002$ |
| NLL | Protein | 41157 | 9 | $0.960 \pm 0.003$ | $\mathbf{0.992 \pm 0.017}$ | $1.213 \pm 0.020$ | $1.406 \pm 0.048$ |
| | Houseelectric | 1844352 | 11 | $0.207 \pm 0.001$ | $\mathbf{0.251 \pm 0.005}$ | $--$ | $0.756 \pm 0.075$ |
| | Elevators | 14939 | 18 | $0.626 \pm 0.043$ | $\mathbf{0.372 \pm 0.004}$ | $0.869 \pm 0.074$ | $1.600 \pm 0.020$ |
| | Keggdirected | 43944 | 20 | $0.838 \pm 0.031$ | $\mathbf{0.254 \pm 0.156}$ | $0.996 \pm 0.013$ | $0.797 \pm 0.031$ |

Table 11: Comparison with SKI-based methods. Numbers for Exact, SKIP and Simplex-SKI are taken from (Kapoor et al., 2021). Best approximate GP numbers are bolded.

To compare against these methods, we attempt to replicate the experimental settings reported by (Kapoor et al., 2021) so that we can directly compare against their reported numbers. The reason for doing so is because Simplex-SKI relies on custom Cuda kernels for an efficient implementation. Consequently, it is difficult to replicate on more current hardware and software stack as both the underlying GPU architectures and PyTorch have evolved. From this perspective, we view the higher-level PyTorch and GPyTorch implementation of SoftKI as one practical strength since it can rely on these frameworks to abstract away these details.

In the original Simplex-SKI experiments, the methods are tested on 4/9, 2/9, and 4/9 splits for training, validation, and testing respectively. For SoftKI, we simply train on 4/9 of the dataset, discard the validation set, and test on the rest. We report the test RMSE and test NLL achieved in Table 11. We find that SoftKI is competitive with Simplex-SKI.

## C.4 Lengthscale and Temperature

Figure 5 gives the histograms of the lengthscales learned by various methods in the experimention in Section 4.1. Figure 6 gives the histogram of the temperatures learned by SoftKI. On datasets such as song and slice, we see that the majority of SoftKI's lengthscales for each dimension reach the maximum lengthscale of 5. The dimensionality-specific variation is instead captured in the temperature.

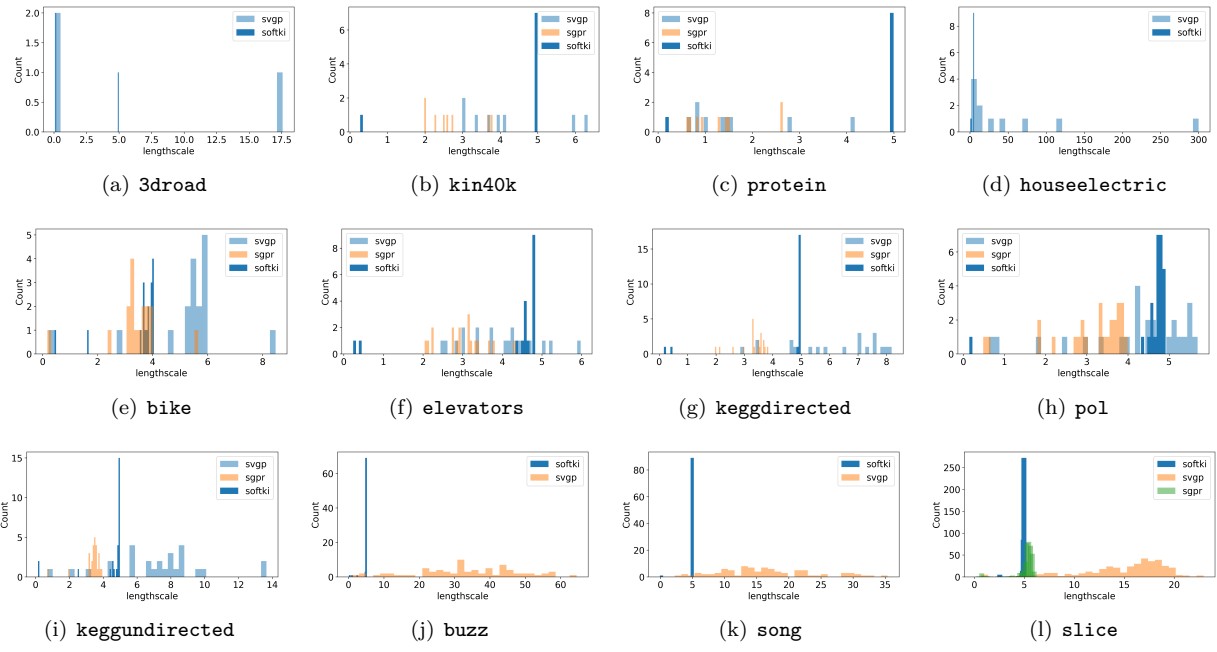

Figure 5: Histogram of lengthscale distributions learned by automatic relevance detection (ARD) by different approximate gaussian process methods on the `uci` dataset.

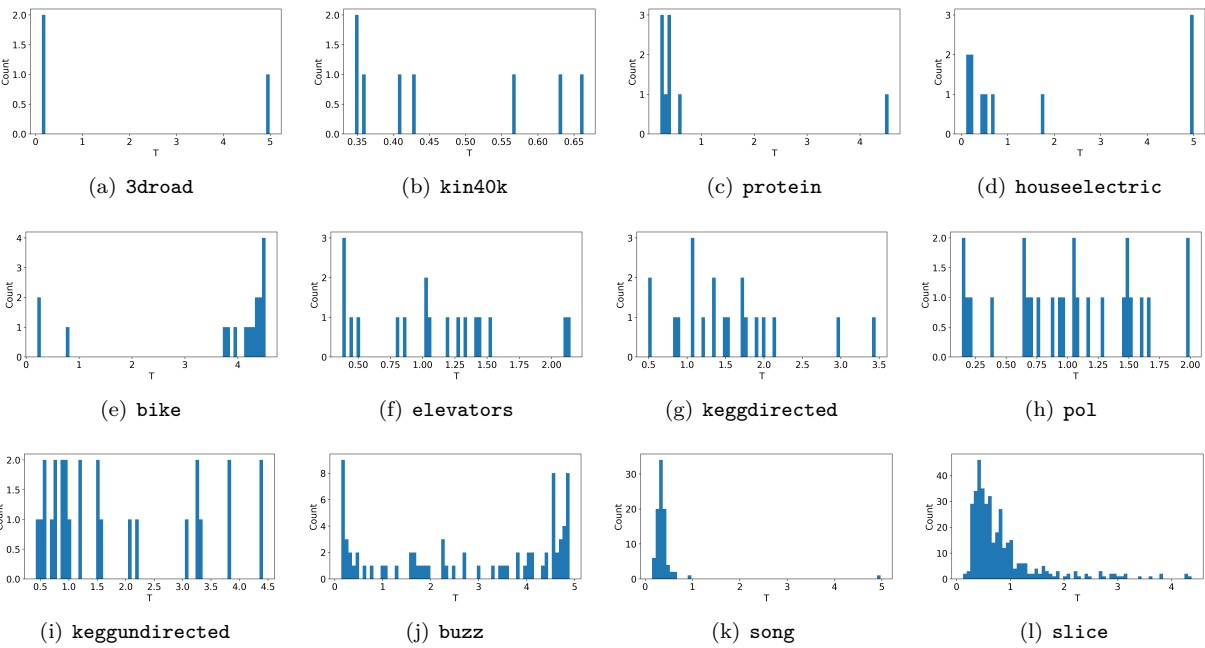

Figure 6: Histogram of Temperatures $T$ learned by SoftKI on the `uci` dataset.

