# OpenReview forum: "High-Dimensional Gaussian Process Regression with Soft Kernel Interpolation"
_TMLR — Accepted by TMLR_

### Review · Reviewer_GRfH · 2025-06-10

**Summary Of Contributions:**

This paper extends the structured kernel interpolation for GP kernels by replacing the sparse matrix W with a softmax matrix and then learning the location of the inducing points. As the resulting GP is low rank, the numerics of the GP matter so the authors use the Hutchinson’s pseudo-loss (from Maddox et al, ’22, Wenger et al, ’22) when the gradients are unstable. For predictive means and covariances, due to the degeneracy, the authors also use a QR decomposition for the solve instead of conjugate gradients.

**Audience:**

Yes

**Claims And Evidence:**

Yes

**Requested Changes:**

Questions:

-	How is the change in the value of the loss dealt with on the optimizer side when the loss used flips to the Hutchinson’s pseudo loss?

-	Have the authors tried directly optimizing the pseudo loss for the entire trajectory (like in Maddox et al, ’22)? As the gradients are the same then the optimization results should be identical, up to numerical differences?

-	SKIP and then Pleiss et al, ’18 (https://arxiv.org/abs/1803.06058) give efficient methods for predictive variances (more so than the algorithm described in Appendix A). Could these methods be applied to speeding up predictive variance computations (e.g. for NLLs) here?

-	Tables 1, 2, 3 and 4: why not include comparisons with exact GPs here (on at least the <50k data sets)? For Tables 1 and 2, I believe that these are basically the same experiments as in Wang et al, ’20, Gardner et al, ’19, and Maddox et al, ’22 (for the floating point experiments).

-	In some cases, the pivoted Cholesky pivots (essentially) (Burt et al, ’20, JMLR[ https://www.jmlr.org/papers/v21/19-1015.html]) are better at initializing GPs in the inducing point setting, did the authors try that instead of k means?

-	Following up from the connection with Yadav et al, ’21, is there a more numerically stable way to compute the predictive mean using the Bayesian linear regression formulation?

-	As this approximation ends up being really subset of regressors (Foster et al, ’09; Smola & Bartlett, ’00), why not compare to just that technique with both fixed and learned inducing points?

**Strengths And Weaknesses:**

Strengths:

-	I appreciate Sections C.3 and C.4 as I understand that the complexity of the algorithms has made some amount of exact reproduction impossible over the course of 3+ years.

-	I like the molecule experiments and the heavy interpretation of them, which are a nice change from just the traditional UCI experiments.

-      Overall, the technique is pretty straightforward and well explained. Good job.

Weaknesses:

-	While I appreciate all of the numerical advances in this paper, I strongly suspect that the reason why the numerics are such a pain under the hood is the fundamental degeneracy of the model itself. The subset of regressor approximations can be viewed as finite rank GPs or Bayesian linear models with |z| features (see Yadav et al, ’21 [https://proceedings.mlr.press/v130/yadav21a/yadav21a.pdf] for a modern description).

-	There’s not a lot of comparisons to exact GPs on the datasets where they can scale reasonably well (see below).

---

> ### Author Response · Authors · 2025-06-24
> **Official reponse to reviewer GRfH**
>
> Thank you for your review. Below are responses to the questions you have raised.
> > How is the change in the value of the loss dealt with on the optimizer side when the loss used flips to the Hutchinson's pseudoloss?
>
> For each minibatch of data, we first calculate the gradient of the MLL. If it is unstable, we instead calculate the gradient of the pseudoloss. Given the gradient (either from the MLL or pseudoloss), we then compute a gradient update step.
>
> > Have the authors tried directly optimizing the pseudoloss for the entire trajectory (as in Maddox et al. I '22)? Since the gradients are identical, would the optimization results differ only by numerical noise?
>
> Prior to developing the Hutchinson pseduoloss fallback logic detailed in equation 17, we experimented running the model using the Hutchinson pseudoloss the entire time. In our experience this approximation comes with a non-negligible accuracy penalty which motivated the hybrid approach currently advocated for in the main text. Additional experiments detailing the effect of running just the Hutchinson Pseduoloss  on the UCI dataset can be found in Appendix B.1.
>
> > Pleiss et al. 18 https://arxiv.org/abs/1803.06058) propose efficient predictive-variance methods in Appendix A. Could these be applied here to speed up variance computations (e.g. I for NLLs)?
>
> Pleiss et al. achieve $\mathcal{O}(k(n+m \log m))$ precomputation and $\mathcal{O}(k)$ per-test-point predictive-variance evaluations by exploiting the multilevel-Toeplitz structure of $K_{z z}$ via FFTs and sparse interpolation weights. In SoftKI, however, the inner kernel does not admit fast matrix-vector products: each application of $K_{z z}$ costs $\mathcal{O}\left(m^2\right)$. As a result, the Lanczos precomputation scales as $\mathcal{O}\left(k\left(n+m^2\right)\right)$, and each subsequent predictive variance evaluation requires $\mathcal{O}(m k)$ rather than $\mathcal{O}(k)$. Additional details can be found in Pleiss et al. '18 Section 3.1 where precomputation for repeated predictive covariance computation is discussed in the context of inducing point methods.
>
> > Why not include comparisons with exact GPs on datasets up to 50k points? For Tables 1 and 2, these experiments resemble those in Wang et al.\'20, Gardner et al.\'19, and Maddox et al.\'22.
>
> In general, scaling exact gaussian process regression to large datasets is challenging. While Gardner et al.  leveraged an eight-GPU cluster and extensive software optimizations to apply exact GPs to the full UCI repository, replicating such an infrastructure was beyond our computational budget for this work. To this end, we chose to highlight the comparison to other approximate gaussian process methods to highlight SoftKI's placement in the broader context of competing algorithms. For datasets that can be stored on a single GPU we are happy to add additional numerics the appendix drawing comparison to exact GPs.
>
> > Pivoted Cholesky initialization (Burt et al.\'20, JMLR) can outperform $k$-means for inducing points. Did the authors compare against that?
>
> We did not include partially pivoted Cholesky for initializing inducing locations. Given Burt et al.'s findings, it is plausible that this strategy could further improve SoftKI's predictive performance. A systematic comparison of Cholesky based, $k$-means, and other bespoke initializations to be a valuable direction for future work.
>
> >Given the Bayesian linear regression view (Yadav et al.I'21), is there a more numerically stable way to compute the predictive mean?
>
> GSGP (Yadav et al.\'21) relies on an $\mathcal{O}(n)$ preprocessing step for fixed interpolation locations, which suits SKI's static grid. In SoftKI, inducing inputs are learned dynamically, so it is not immediately clear how to amortize this preprocessing efficiently. Adapting GSGP's linear-time updates to a learned-inducing-point regime presents an interesting avenue for further investigation.
>
> >Since this method reduces to subset of regressors (Foster et al.\'09; Smola \& Bartlett '00), why not compare directly to that technique with fixed and learned inducing points?
>
> Although SoftKI's objective shares similarities with subset-of-regressors, SOR yields a degenerate (rank-deficient) posterior covariance. By contrast, SoftKI maintains a full-rank covariance (see equation~18 and Appendix A), ensuring nondegeneracy in predictive uncertainty. For this reason, we did not include SOR as a direct baseline.

---

> > ### Comment · Reviewer_GRfH · 2025-07-08
> > **thanks for the comments**
> >
> > Thanks overall for the comments, and replies to my suggestions / comparisons.
> >
> > >  In our experience this approximation comes with a non-negligible accuracy penalty
> >
> > If the gradients are the same up to numerical precision, this can only come from operations done on the forwards pass of the loss (i.e. the optimizer stopping rule). Thus, i find it surprising that there is in fact an accuracy penalty. A simple comparison might be to run one of the smaller datasets in _double_ precision for all three approaches (full lml optimization, full pseudoloss, switching) to see what's going on.
> >
> > >  If it is unstable, we instead calculate the gradient of the pseudoloss.
> >
> > What exactly does "unstable" mean here? is this the solve not diverging?
> >
> > > Pleiss et al comparisons
> >
> > Indeed, that makes sense, and thank you for reminding me of the fast MVMs that generally makes it work. I believe that the predictive variance decomposition they are doing is probably a bit faster and more stable than yours even if yours is technically a bit more full rank.
> >
> > > SOR yields a degenerate (rank-deficient) posterior covariance.
> >
> > Note that in some settings SKI can also produce a rank-deficient posterior covariance (e.g. if the number of inducing points is less than the number of data points...). Thus, I think it might be a reasonably fair comparison.
> >
> > > For datasets that can be stored on a single GPU we are happy to add additional numerics the appendix drawing comparison to exact GPs.
> >
> > I believe that all of the comparisons from Maddox et al, '22 are on a single GPU for example; and specifically the <=50k datasets should be very doable. Note that generally larger GPUs (ie H100s) would allow for another doubling of capacity...
> >
> > > GSGP (Yadav et al.'21) relies on an $\mathcal{O}(n)$ preprocessing step for fixed interpolation locations
> >
> > I wasn't really pointing out the connection to "cheap" online learning per se, although Stanton et al, '21 (https://arxiv.org/pdf/2103.01454) do actually perform something similar to what you proposed there (update the inducing points / deep learning head) in batch..
> >
> > What I meant more specifically is that the specific SKI-style inducing point approximation boils down to Bayesian linear regression (see Definition 2 in their paper), which might give rise to a more stable predictive mean. In your case, this is empirical Bayesian as you're learning the prior covariance matrix alongside things while optimizing the log marginal likelihood / pseudo loss.

---

> > > ### Comment · Reviewer_GRfH · 2025-07-08
> > > **other comment**
> > >
> > > > We did not include partially pivoted Cholesky for initializing inducing locations.
> > >
> > > I should also note that this may end up including a bit more repulsion / spread (and thus stability compared to k means) in the inducing point optimization as Burt et al, '20 point out that this initialization scheme ends up looking a lot like repulsive determinental point processes.

---

> > > ### Author Response · Authors · 2025-07-10
> > > **Response to additional comments.**
> > >
> > > >For datasets that can be stored on a single GPU we are happy to add additional numerics... I believe that all of the comparisons from Maddox et al, '22 are on a single GPU for example;
> > >
> > > Thank you for the feedback, as a point of comparison, we ran exact Gaussian process regression on the UCI datasets featured in Maddox et al '22. Below are the test RMSE results averaged over three runs following the kernel initialization procedure detailed in Maddox et all '22 as closely as possible.
> > >
> > > |                  |        N |   d | softki        | exact         |
> > > |:--------------|--------:|----:|:--------------|:--------------|
> > > | 3droad        |  391,386 |   3 | 0.573 ± 0.007 | 0.194 ± 0.01  |
> > > | Kin40k        |   36,000 |   8 | 0.17 ± 0.002  | 0.099 ± 0.001 |
> > > | Protein       |   41,157 |   9 | 0.623 ± 0.053 | 0.055 ± 0.006 |
> > > | Houseelectric | 1,844,352 |  11 | 0.045 ± 0.0   | 0.052 ± 0.002 |
> > > | Bike          |   15,641 |  17 | 0.047 ± 0.004 | 0.074 ± 0.003 |
> > > | Elevators     |   14,939 |  18 | 0.361 ± 0.007 | 0.364 ± 0.004 |
> > > | Pol           |   13,500 |  26 | 0.076 ± 0.002 | 0.117 ± 0.004 |
> > > | Buzz          |  524,925 |  77 | 0.238 ± 0.001 | 0.300 ± 0.01  |
> > > | Song          |  270,000 |  90 | 0.777 ± 0.004 | 0.761 ± 0.004 |
> > >
> > > These details will be added to the appendix of the main text.
> > >
> > > > Note that in some settings SKI can also produce a rank-deficient posterior covariance.  Thus, I think it might be a reasonably fair comparison.
> > >
> > > Because the interpolation matrix $\mathbf{W}_{\mathbf{xz}}$ used in SoftKI's kernel approximation is formed by a dense, global softmax interpolation rather than the locally supported cubic-interpolation used in SKI, we agree that SoftKI naturally fits into the family of global kernel interpolation schemes such as SOR and its improved relative FITC. Understanding the positioning of SoftKI in the hierarchy of these other global interpolation methods is likely a theoretically fruitful direction since it may be possible to re-derive statements about SoftKI using known results for FITC or SOR. For the scope of this work we chose to limit our comparison to other leading methods for approximate gaussian process regression as an empirical demonstration of the methods performance. We are happy to add emphasis to this comparison to the discussion of related work.
> > >
> > > > What exactly does "unstable" mean here? is this the solve not diverging?, A simple comparison might be to run one of the smaller datasets in double precision for all three approaches (full lml optimization, full pseudoloss, switching) to see what's going on.
> > >
> > > Yes, by *unstable* we mean situations in which the marginal log likehood computation fails due to the inducing point kernel $\mathbf{K}_{\mathbf{zz}}$ failing to be positive semidefinite in floating point precision. To address this, we use the `gpytorch.linear_operator` routine `psd_safe_cholesky`, which performs a stabilized Cholesky factorization by iteratively adding increasing large perturbations to the diagonal until successful. If this method fails we then invoke the Hutchinson pseduloss as a last resort. Attached is the requested example comparing a subset of the UCI dataset in float64 using
> > > - Exact mll computation (softki-exact )
> > > -  Hutchinson pseudoloss only  (softki-hutchinson)
> > > - The combination strategy described in the paper (softki-hutchinson_fallback )
> > >
> > > |                |       N |   D | softki-exact   | softki-hutchinson   | softki-hutchinson_fallback   |
> > > |:---------------|--------:|----:|:---------------|:--------------------|:-----------------------------|
> > > | Pol            |   13500 |  26 | 0.077          | 0.084               | 0.077                        |
> > > | Elevators      |   14939 |  18 | 0.356          | 0.357               | 0.356                        |
> > > | Bike           |   15641 |  17 | 0.044          | 0.075               | 0.044                        |
> > >
> > > In float64 $\mathbf{K}_{\mathbf{zz}}$ is safer to work with, and we see that the combination strategy recovers the exact computation in all three examples.
> > >
> > > > What I meant more specifically is that the specific SKI-style inducing point approximation boils down to Bayesian linear regression (see Definition 2 in their paper), which might give rise to a more stable predictive mean
> > >
> > > Thank you for clarifying, Definition 2 of the Yadav et al 21' paper presents a formalization of interpolative approximate gaussian processes (GSGP). In general, the purpose of the GSGP is to reframe SKI as a bayesian linear regression task such that approximate kernel inversion via CG is more efficient ($\mathcal{O}(n)+\mathcal{O}(pm\log m)$ vs $\mathcal{O}(p(n4^d+m\log m)$ for p iterations).  While there are structural similarities between SoftKI and GSGP, a GSGP requires the underlying grid to be structured so it is not clear that this abstraction would generalize naturally to SoftKI. We are happy to mention this work as well in the related work (Section 3) as it is a very relevant comparison.

---

> > > > ### Comment · Reviewer_GRfH · 2025-07-23
> > > > **last comment here**
> > > >
> > > > comparisons:
> > > >
> > > > Thanks for the comparisons between the three approaches; I'm happy that you ran them. However, I'm quite confused as the optimization trajectory of all three should be exactly the same up to numerical precision due to the backwards being the same. So, I find it quite strange that the downstream performance is so different... The only difference I can imagine is the forwards pass of both versions being different, and that changing the stopping rule of the optimizer somehow

---

> > > > > ### Author Response · Authors · 2025-07-23
> > > > >
> > > > > Thank you for your response. The forward pass when Hutchinson's pseudoloss (distinguished with notation $\log \tilde{p}(y | x; \theta)$ in the text) is used is an approximation of the MLL based on an empirical average whose accuracy is dependent on the number of probes. We will clarify in the main text that it is an approximation of the MLL so that the gradients will also be approximated.

---

> > > > > > ### Comment · Reviewer_GRfH · 2025-07-23
> > > > > >
> > > > > > Yes, that makes sense. However, the backwards of the "approximation" is the same as the approximation in https://arxiv.org/pdf/1809.11165 (see Eqs 4 and 5 there as well as Eqs 2 and 8 in https://arxiv.org/pdf/2207.06856). The dropoff in performance then is probably due to any stochastic trace estimate based approach vs the exact gradient from cholesky, not the hutchinson pseudo-loss.

---

> > > > > > > ### Author Response · Authors · 2025-07-23
> > > > > > >
> > > > > > > Yes, we agree with your assessment that the dropoff in performance is due to the stochastic trace estimate based approach vs. an exact gradient. We will clarify in the main text that the gradient of the Hutchinson pseudoloss is an approximation of the gradient of the MLL where Hutchinson's stochastic trace estimator is used.

---

### Review · Reviewer_GuQj · 2025-06-11

**Summary Of Contributions:**

The contribution presents a new method in the SKI family where rather than exploiting sparsity and structure in the interpolation matrices of points selected a priori, it learns the interpolation points using an approximate sparsity constraint. While this brings back quadratic dependence on the number of interpolation points, it removes the dependence on data dimensionality, which allows the method to scale to higher-dimensional datasets. Further, the fact that interpolation points are learned lets the approach perform well on these datasets even without requiring very many interpolation points. A number of additional techniques are used to make inference stable and tractable.

**Audience:**

Yes

**Broader Impact Concerns:**

No concerns.

**Claims And Evidence:**

Yes

**Requested Changes:**

See discussion above -- I think my main requested change is addressing the settings of high $m$ low $d$ vs low $m$ high $d$ and trying to situate the new contribution in a way that is less confusing for readers. I also strongly encourage addressing all the minor points. I think an additional baseline or two against contemporary methods with favorable scaling would also be beneficial but I don't think it's critical, especially given the TMLR rubric.

**Strengths And Weaknesses:**

# Strengths:
The paper pursues a new branch in the SKI family (low-rank rather than sparse structured interpolation points), and gets it to work using a combination of contemporary tricks -- it looks like a solid contribution and a new model that should be in the practitioner's toolbox. The paper is well written, notation is clear, and benchmarks are extensive and show the method in favorable light. The new idea (learned softmax interpolation points) is potentially applicable alongside other structured SKI tricks to increase scale further.

# Weaknesses:

I think the paper is good overall -- I have two bigger comments on baselines and potentially confusing exposition, plus some minor points and suggestions.

## The $m \ll n$ (high $d$) vs $m \gg n$ (low $d$) setting

I was confused during my initial read of the paper because I kept looking for the part where SoftKI admits fast matmuls (as other SKI methods do) -- but that's not the point here, the "fast" matmul is simply matrix inversion lemma on a low-rank interpolation points matrix. My read on the various SKI methods is that they operate in the large-$m$ setting by trading off very good scaling in $m$ against unfavorable scaling in $d$, and attempt to come up with clever fixed interpolation schemes that make the scaling in $d$ less unfavorable. In contrast, the present approach accepts worse scaling in $m$ (by eschewing structure in the interpolation points) and removes the dependence on $d$ (by eschewing fixed interpolation schemes). Is that right? If so, this is all great, but it took me a few reads and going back to the SKI papers to realize this is what was going on, and making it explicit would help a reader (e.g. by reminding us that inducing point approaches often take $m \ll n$ but SKI approaches can take $m \approx n$ or even $m \gg n$, or that it takes a different branch from other SKI approaches). It's all in there in the paper, and once I figured out what's going on I saw it, but better signposts would have helped. One possible idea is just a table with methods and their scaling in $m, n$, and $d$ which would make it clearer that the present contribution is an SKI-like method with SGPR-like scaling.

## Comparing against other methods

SGPR and SVGP are good "reference" baselines, and it's also nice to see the favorable behavior relative to Skip and Simplex-SKI in the supplement, but there are other GP methods that do scale to the sizes used in this contribution. For example, VNNGP (which the present paper cites) has linear scaling in both $n$ and $m$ and no explicit dependence on $d$ (though the nearest-neighbor algorithm incurs a one-time cost dependent on $d$), and as such can operate on a n=372K d=54 dataset which is not far off from what is studied here. GPnn (Allison et al. 2023) has similar scaling and tests on datasets with $d$ up to 378. CaGP (Wenger et al. 2024) has favorable scaling in both $m$ and $d$ and runs experiments on datasets with n up to 1.8M. I don't expect the present contribution to compare against all of them (and under the TMLR rubric, I don't think comparison against *any* of them is needed to make the point that the present contribution is correct and interesting), but it would strengthen the paper, particularly because my sense is that SoftKI may implicitly find a set of neighbors for each point during optimization. It would also make the comparison to where the interpolation points are (section 5.3) potentially more interesting.

## Minor points
* SKI vs KISS-GP: Wilson and Nickisch make a distinction between SKI (the approximation given in expr [7] in the present paper) and KISS-GP (SKI with cubic interpolation and with structure-exploiting linear algebra). The review in section 2.3 does not make that distinction, and only reports the complexity of the Toeplitz approach, whereas the KISS-GP paper also discusses Kronecker.
* It's true that naive GP inference is $O(n^3)$ complexity (as claimed in the introduction) but I think by now the fast MVM iterative solver approach has been around long enough to where we can claim $O(n^2)$ is the baseline for modestly large $n$. I don't think there's a need to background this under "efforts have been made to accelerate exact GP inference" especially considering that those tricks underpin the standard SKI approach.
* The notation for a Gaussian distribution ($\mathcal{N}(\cdot \mid \mu, \mathbf{\Sigma})$) is defined after expression 2 even though it is used in the model likelihood and expr 1 above. So it should possibly be moved up, though I'm also not sure that it needs to be defined for an ML audience.
* Fig 1 caption should explicitly describe what the magenta segments are (they are connecting the points used for interpolation?).
* I'm not sure what to make of Fig 3 and section 5.3. It looks like the distribution of SoftKI points doesn't overlap as much with the data over the first two components as SGPR / SVGP, the conclusion isn't much beyond "they're different" and the discussion is vague. I think it could easily be moved to the appendix in favor of bringing up the Skip and Simplex-SKI results.

---

> ### Author Response · Authors · 2025-06-24
> **Official reponse to reviewer GuQj**
>
> Thank you for your review and thoughts on the work. We agree with all of the minor points you suggest and will make adjustments accordingly to reflect these changes in the writing of the main text. For clarification in Figure 1, the magenta segments highlight exactly which interpolation points are chosen for each data sample. In the case of SKI it's the three closest in the grid, whereas for SoftKI its all points with importance weighted by a softmax distribution. We will revise the caption to make this more explicit. We also feel that the comparison to SKIP and Simplex SKI may be a more natural discussion to have in place of the PCA analysis in section 5.3.
>
> - **The $m \ll n$ (high $d$ ) vs $m \gg n$ (low $d$ ) setting**
>
> Thank you for sharing your reading experience. You are exactly right in your final conclusions about the proposed changes in SoftKI, and we agree that our current exposition could more clearly emphasize the perspective shift taken by SoftKI. Including a summary table of each method’s $(n,m,d)$-dependence is an excellent suggestion. We will add such a table and revise the surrounding discussion to leave a clearer trail for the reader that draws comparisons between the high-d $(m\ll n)$ (high $d$) to the  $(m\gg n)$ (low $d$) regime.

---

### Review · Reviewer_HaUc · 2025-06-17

**Summary Of Contributions:**

The authors develop a variation of the kernel interpolation approach to approximate a GP called SoftKI. SoftKI chooses quadrature points via optimization and assigns the quadrature weights based on a softmax over distances. The method is integrated with standard techniques from variational inference to allow for batched optimization and is tested on well-known datasets.

**Audience:**

Yes

**Broader Impact Concerns:**

I have no concerns about a negative broader impact.

**Claims And Evidence:**

Yes

**Requested Changes:**

### Major Issues
- The method section largely reads like a laundry list of what needs to be done to make the method work. While those are all important points, more detailed derivations, especially on SoftKI and the stabilized MLL, are IMO in order. In its current form, the paper resembles a successful empirical study. With supporting theory, there might be subsequent work building on it. In particular
  - "4.2 Learning Interpolation Points"
  - (10)
  - (17)
- p.7 "when numeric instability is encountered", state the algorithm explicitly
- The Nystroem method approximates the posterior. We approximate the Kernel with Methods like RFF. Before truncating, those representations are equivalent, see [2], after truncating, they are different and induce different approximation artifacts (variance starvation, loss of resolution).

### Minor Issues
- Figure 1 needs more interpretation. The distribution of inducing points is explained, the distribution of weights needs more dissemination.
- The ELBO is never stated in its generic form; it might be good for the exposition to include it in (6).
- "2 Background" and "3 Related Work" have no clear distinction.
- For the Figures with multiple axes, please use subcaptions instead of the legend to label the axes.
- Table 6, in the last row "Slice", is the wrong number bolded?
- SGPR is still in the table in spite of "We do not include SGPR"
- I cannot follow the line of thinking here: p.18, "Second, the patterns in the interpolation points learned by SoftKI confirm our intuition that high-dimensional spaces can benefit from fewer interpolation points." Are there concrete examples? I cannot spot a qualitative difference.
- The appendix is not well formatted, eg.: p.19ff trailing text under the figure, missing captions, sometimes the inducing points for SGPR seem to be missing. The captions are, in general, not very descriptive.

### Questions
- The authors perform an ablation wrt. the number of inducing points. Surprisingly, more interpolation points do not lead to better performance in general. They state that this might be due to overfitting. Does the
- The paper puts a lot of emphasis on the low computational complexity of single gradient updates. Yet, the algorithm uses dense matrices. I was wondering why not discuss sampling from the SoftKI? It is explicitly interpreted as
a probability distribution over interpolation points for each datapoint. This could sparsify the gradient and robustify the model in a similar fashion to dropout.
- Plotting inducing points along PC is often used to showcase how SoftKI is "efficient". In isolation, I don't know what is expected to be observed in those plots. Is there some intuition I am missing?


[2]: Bach, Francis. "On the equivalence between kernel quadrature rules and random feature expansions." Journal of machine learning research 18.21 (2017): 1-38.

**Strengths And Weaknesses:**

### Strengths
- The paper has a clear goal of allowing kernel interpolation to scale to datasets with higher-dimensional feature vectors.
- To the best of my knowledge, KISS-like GPs are still the SoTA when it comes to using kernel interpolation in GPs.
- The method is extensively evaluated and compared.

### Weaknesses
- The main idea of the paper (the use of the softmax for quadrature weights in (10)) lacks a derivation or even a strong explanation.
- The paper makes some statements based on intuition; to strengthen the statements and for completeness, these should be supported by a formal argument (proof) or a reference. E.g.:
  - p.6, "they have a push-and-pull effect"
  -  p.8, "The stochastic estimate provides an unbiased estimator of the gradient."
- While the method removes the explicit dependence on the feature dimensionality (by removing the grid structure), it is unclear whether SoftKI can utilize the underlying manifold structure postulated for many practical problems. The efficiency could then be explained by, ie, [1].
- The paper is too empirical. Bayesian inference through GPs is IMO a very useful tool for the ability to derive and explain algorithms.

[1]: Rosa, Paul, et al. "Posterior contraction rates for Matérn Gaussian processes on Riemannian manifolds." Advances in Neural Information Processing Systems 36 (2023): 34087-34121.

---

> ### Author Response · Authors · 2025-06-24
> **Official reponse to reviewer HaUc**
>
> Thank you for your review, we respond to your questions below.
> >The method section largely reads like a laundry list of what needs to be done to make the method work. While those are all important points, more detailed derivations, especially on SoftKI and the stabilized MLL, are IMO in order.
>
> The goal of our work is to develop a new method for large $n$ and large $d$ settings that a practitioner could use. As such, our focus is on finding the right algorithmic ingredients to make the method practical and a subsequent empirical study. We wholeheartedly agree that theory of the method is underexplored, albeit not a focus of the current paper.
>
> > p.7 "when numeric instability is encountered", state the algorithm explicitly
>
> The logic controlling whether to evaluate the exact marginal log-likelihood or switch to the Hutchinson pseudo-loss is described in Equation 17, which  implements a conditional check. In retrospect, this presentation could have been made clearer; we will revise the exposition in future drafts to reflect this. In this situation, we refer to *numerical instability* as the failure of the inducing point kernel $\mathbf{K}_{\mathbf{z}\mathbf{z}}$ to be positive semi-definite (see Appendix B.1 for more details if needed). To address this, we employ the `gpytorch.linear_operator` routine`psd_safe_cholesky`, which performs a stabilized Cholesky factorization by iteratively adding jitter to the diagonal until successful. If this method fails we then invoke the Hutchinson pseduloss as a last resort to overcome this challenge.
>
> >The ELBO is never stated in its generic form; it might be good for the exposition to include it in (6).
>
> Thank you for the suggestion, we agree that this may help the discussion on sparse gaussian process regression and will include it in the revised draft.
>
> > "2 Background" and "3 Related Work" have no clear distinction.
>
> Thank you for the suggestion, we intended background to introduce the basics of GPs, SGPR, and SKI, and related work to introduce additional SKI variants. This can be more clearly communicated by updating the section titles.
>
>  - For the Figures with multiple axes, please use subcaptions instead of the legend to label the axes.}
>
> -  Table 6, in the last row "Slice", is the wrong number bolded?
>
> - SGPR is still in the table in spite of "We do not include SGPR"
>
> We appreciate your detailed feedback on the layout and captions of our figures in both the main text and the appendix, we will revise accordingly. Specifically, we plan to rewrite Figure 1’s caption for greater clarity and to adopt a more conversational tone for all appendix figure captions.
>
> > I cannot follow the line of thinking here: p.18, "Second, the patterns in the interpolation points learned by SoftKI confirm our intuition that high-dimensional spaces can benefit from fewer interpolation points." Are there concrete examples? I cannot spot a qualitative difference.
>
> Thank you for this question. We mean that a PCA of a high-dimensional structured grid would be different from the PCA of the learned structure, indicating that we can use fewer interpolation points. We will clarify this.
>
> > The authors perform an ablation wrt. the number of inducing points. Surprisingly, more interpolation points do not lead to better performance in general. They state that this might be due to overfitting. Does the..
>
> Could you please finish this thought so that we can better respond?
>
>
> >The paper puts a lot of emphasis on the low computational complexity of single gradient updates. Yet, the algorithm uses dense matrices. I was wondering why not discuss sampling from the SoftKI? It is explicitly interpreted as a probability distribution over interpolation points for each datapoint. This could sparsify the gradient and robustify the model in a similar fashion to dropout.}
>
> Thank you for this suggestion! This is an interesting direction of future work which we had hinted at this in our concluding sentence: "Additionally, exploring methods to enforce stricter sparsity in the interpolation matrices..." and will extend to include your suggestion.
>
> > Plotting inducing points along PC is often used to showcase how SoftKI is "efficient". In isolation, I don't know what is expected to be observed in those plots. Is there some intuition I am missing?
>
> The primary purpose of the PCA analysis in the main text was to show that SoftKI selects fundamentally different interpolation point locations compared to the inducing points learned by other variational methods, which aim to capture the underlying data distribution. We intended the visual distinctions to make this contrast immediately clear, but we recognize that the presentation may not have achieved that. In response to `reviewer GRfH` ’s feedback, we will now supplement this section with the Appendix C.3 discussion, which offers a direct comparison to existing SKI variants and Exact GP.

---

> > ### Comment · Reviewer_HaUc · 2025-06-30
> > **Futher questions on SoftKI**
> >
> > Thank you for your detailed response to my questions.
> > >
> > >> The method section largely reads like a laundry list of what needs to be done to make the method work. While those are all important points, more detailed derivations, especially on SoftKI and the stabilized MLL, are IMO in order.
> > >
> > > The goal of our work is to develop a new method for large $n$ and large $d$ settings that a practitioner could use. As such, our focus is on finding the right algorithmic ingredients to make the method practical and a subsequent empirical study. We wholeheartedly agree that theory of the method is underexplored, albeit not a focus of the current paper.
> >
> > This goal is relevant and treated nicely. However, it would greatly strengthen the paper and allow for future work to include at least some intuition and possibly directions for analysis of the SoftKI approximation.
> >
> > >
> > >> I cannot follow the line of thinking here: p.18, "Second, the patterns in the interpolation points learned by SoftKI confirm our intuition that high-dimensional spaces can benefit from fewer interpolation points." Are there concrete examples? I cannot spot a qualitative difference.
> > >
> > > Thank you for this question. We mean that a PCA of a high-dimensional structured grid would be different from the PCA of the learned structure, indicating that we can use fewer interpolation points. We will clarify this.
> >
> > Thank you for the intuition. However, a *different* structure in the PC does not imply a more *efficient* structure. It is not convincing that the proposed inducing point scheme is more efficient. To support your claim could measure this effect directly by measuring a quantity that would reflect this. A measure for a hypothesis space allowing for low-rank approximations is the information gain that leads to efficient learning algorithms [1] and inducing point methods [3] . Since the information gain is equivalent to other information-theoretic quantities, like effective dimension [2][Appendix D], you can compute the most convenient metric (with reasonable but unoptimized hyperparameters) and compare them between your SoftKI (11) and other schemes such as SKI (8) or SGP (3).
> >
> > >
> > >> The authors perform an ablation wrt. the number of inducing points. Surprisingly, more interpolation points do not lead to better performance in general. They state that this might be due to overfitting. Does the..
> > >
> > > Could you please finish this thought so that we can better respond?
> >
> > The last part of my question got cut off when checking for errors. I wanted to ask: 'Does the use of a pseudo loss or SoftKI introduce this effect? The Nystroem method gets unstable as M gets close to N because ill-conditioned matrices need to be inverted, but here, M seems to be close to 10% of N.' In light of reviewer GRfH's comments and my previous comment on the effective dimension, maybe this has to do with a very low-rank Kzz in (11)?
> >
> > [1]: Srinivas, Niranjan, Andreas Krause, Sham M. Kakade, and Matthias W. Seeger. 2012. “Information-Theoretic Regret Bounds for Gaussian Process Optimization in the Bandit Setting.” IEEE Transactions on Information Theory 58 (5): 3250–65. https://doi.org/10.1109/TIT.2011.2182033.
> >
> > [2]: Zenati, Houssam, Alberto Bietti, Eustache Diemert, Julien Mairal, Matthieu Martin, and Pierre Gaillard. 2022. “Efficient Kernelized UCB for Contextual Bandits.” In Proceedings of The 25th International Conference on Artificial Intelligence and Statistics, 5689–5720. PMLR. https://proceedings.mlr.press/v151/zenati22a.html.
> >
> > [3]: Burt, David R., Carl Edward Rasmussen, and Mark van der Wilk. 2020. “Convergence of Sparse Variational Inference in Gaussian Processes Regression.” Journal of Machine Learning Research 21 (131): 1–63.

---

> > > ### Author Response · Authors · 2025-07-06
> > > **Response to further questions on SoftKI**
> > >
> > > > This goal is relevant and treated nicely. However, it would greatly strengthen the paper and allow for future work to include at least some intuition and possibly directions for analysis of the SoftKI approximation.
> > >
> > > Thank you for the feedback, during the development of the project we considered multiple avenues one could potentially consider for building out a richer theoretical characterization of the method. One promising direction is extending existing results for greedy kernel interpolation methods [1] (see [2] for a more modern treatment in the general setting) to SoftKI. Given interpolation points $\mathbf{z}\subset\mathbb{R}^n$ and data points $\subset\mathbb{R}^n$, consider the RKHS power function
> > > $$P_\mathbf{z}(x)
> > > =\sqrt{k(x,x)-k(x,\mathbf{z})\,K_{\mathbf{z}\mathbf{z}}^{-1}\,k(\mathbf{z},x)}$$
> > > where $k(x,\mathbf{z})=[k(x,z_j)]$ $j=1,\ldots, m.$ For the Gaussian RBF one can show by $1-e^{-t}\le t$ that
> > > $P_\mathbf{z}(x)\le\frac{1}{\ell}\min_{1\le j\le m}\||x-z_j\||.$
> > > SoftKI then constructs the softmax‐interpolation weights
> > > $$\Sigma_{\mathbf{x}\mathbf{z}}[i,j]
> > > =\frac{\exp\bigl(-\||x_i-z_j\||/T\bigr)}
> > > {\sum_{k=1}^m\exp\bigl(-\||x_i-z_k\||/T\bigr)},$$
> > > Setting $\beta=\ell/T$ (as is done in our method) gives
> > > $\Sigma_{\mathbf{x}\mathbf{z}}[i,j]\propto\exp\bigl(\beta\,P_\mathbf{z}(x_i)\bigr)$
> > > i.e. a Boltzmann distribution on the power‐function values. Adapting the existing convergence proofs for P-greedy or other related  greedy selection algorithms to SoftKI by studying the effect of the intermediate softmax sampling on the power‐function is likely a fruitful route towards understanding the convergence properties of the method. We will update the main text to leave a stronger trail for future follow-up work concerned with the theoretical properties of SoftKI.
> > >
> > > [1] Marchi, Schaback and Wendland, Near-optimal data-independent point locations for radial basis function interpolation
> > >
> > > [2] Albrecht, Iske, On the convergence of generalized kernel-based interpolation by greedy data selection algorithms
> > >
> > > > Thank you for the intuition. However, a different structure in the PC does not imply a more efficient structure. It is not convincing that the proposed inducing point scheme is more efficient....
> > >
> > > We agree that PC does not fully justify the effectiveness of SoftKI. In general the purpose of this experiment was to demonstrate the qualitative difference between inducing points and the interpolation points of SoftKI. Following your suggestion, we decided to investigate information gain as an alternative metric of inducing point efficiency. We compute the information gain as described in [1]: Seeger. et al 2012. “Information-Theoretic Regret Bounds for Gaussian Process Optimization in the Bandit Setting.”  Section B implemented as
> > >
> > > $$
> > > \mathcal{I}(\mathbf{y} ; f)=\frac{1}{2} \log \operatorname{det}\left(I+\frac{1}{\sigma^2} \tilde{K}_{\mathbf{x x}}\right)
> > > $$
> > >
> > > where  $\tilde{K}_{\mathbf{x x}}$ is chosen to be the kernel approximation of a SGPR, SVGP, or SoftKI. We use the same parameter settings for the numerical experiments in the main text and perform no hyper parameter optimization using the same inducing point initialization for all methods. Below are the results for UCI datasets considered in this work that fit on a GPU with 12GB of RAM.
> > >
> > > | Dataset    |     n |  d | SGPR IG (±)           | SVGP IG (±)           | softKI IG (±)           |
> > > |------------|-------|----|------------------------|------------------------|--------------------------|
> > > | bike       | 15641 | 17 | 87.1255 ± 1.0838        | 70.8490 ± 0.9619        | **160.5962 ± 4.8726**     |
> > > | elevators  | 14939 | 18 | 86.1461 ± 2.0094        | 70.5805 ± 1.8474        | **103.2958 ± 3.4930**     |
> > > | pol        | 13500 | 26 | **125.0559 ± 4.4087**   | 107.3004 ± 4.1515       | 43.4203 ± 2.2949          |
> > >
> > > Among the three examples able to be tested on hardware currently available to us, in most cases SoftKI is competitive with the alternative methods. As a point of reference SoftKI is the most performant method in terms of both test RMSE and NLL for all three of these datasets after training.
> > >
> > > >  'Does the use of a pseudo loss or SoftKI introduce this effect?...
> > >
> > > At present we do not suspect the cause for this effect to be an consequence of rank degeneracy issues in $ \mathbf{K}_{\mathbf{zz}} $ introducing increased dependence on the hutchsinon pseuduoloss as the approximation is typically quite accurate. One possible explanation is that the current model does not feature a repulsive term between interpolation points. As such, if a surplus of interpolation points are used, it is possible that they may collapse into the same local neighborhood in high dimensions leading to overfitting. Investigating a modification of the model that ensures this effect does not occur may potentially allow the model to scale more aggressively in the number of interpolation points.

---

### Decision · Action_Editor_JMJ5 · 2025-07-26

**Recommendation:** Accept with minor revision

**Additional Comments:**

Overall this paper provides a novel method backed by comprehensive experimental evidence of its efficacy. The paper is well-written and of interest to the GP community and TMLR audience.

The authors responded constructively to reviewer concerns and adequately addressed outstanding issues. However, their promised revisions have not yet been incorporated into the manuscript. I am conditionally accepting the paper, subject to the authors submitting a revision implementing the promised changes, including:

- Non-GP specific version of ELBO
- Comparisons to exact GPs
- Ablation study on optimization numerics (exact MLL vs Hutchinson pseudoloss)
- Discussion of stability issues in SoftKI and other low-rank GP approximations
- Explicit text or table detailing SoftKI's scaling/computational complexity

In addition, I would like to see whether the numerical stability methods proposed by the authors provide benefit to other GP approximations (if applicable). These experiments would provide evidence towards whether or not SoftKI is more numerically unstable than other scalable GP methods.

**Audience:**

Yes

**Audience Explanation:**

This method provides another tool in the scalable GP toolbox that complements methods like structure kernel interpolation and inducing point methods. It will be of interest to members of the TMLR audience who are interested in new scalable GP algorithms.

**Claims And Evidence:**

Yes

**Claims Explanation:**

The paper introduces a new method—soft kernel interpolation—which approximates a full kernel matrix by applying softmax interpolation to an inducing point kernel matrix. The authors propose numerically stable methods for optimizing hyperparameters and computing posterior distributions, borrowing ideas from previous methods (e.g., the Hutchinson pseudoloss).

The description of the proposed method is very clear, though runtime/complexity analysis is difficult to discern from the paper.
There is little theoretical justification, but that is not necessary for "accurate, convincing, and clear evidence."
There are extensive experiments across multiple settings, ranging from accuracy results to visualizations and ablation studies on numerical method choices. These experiments show that the method is efficacious in various settings and can serve as a useful alternative to existing methods (the main proposition of the paper).

One point that could benefit from more evidence is analysis of the algorithm's numerical stability. While the authors provide many methods to prevent instability during training and inference, it remains unclear whether this method suffers from numerical issues more than other scalable methods. Evidence addressing this point would be useful (see below).

---

> ### Author Response · Authors · 2025-08-16
> **Response to Editor**
>
> Hello and thank you for the feedback we have addressed your editorial requests in the camera ready of the draft. For completeness we have documented the changes below.
>
>
> > Non-GP specific version of ELBO
>
> Following this comment (and reviewer HaUc) we have reformatted the section on SGPR to more clearly motivate the ELBO objective, presenting a general form before specializing it to the kernel approximation produces by SGPR.
>
> > Comparisons to exact GPs
>
> We have systematically integrated a comparison to ExactGP (via \texttt{gpytorch} ) into all numerical experiments in section 5. The results of ExactGP can be found as a comparison on the left of the test RMSE and test NLL produced by other approximate GP. For configuration settings we used \texttt{keops} to scale ExactGPs to large datasets and also report per epoch runtime statistics in the appendix for comparison.
>
> > Ablation study on optimization numerics (exact MLL vs Hutchinson pseudoloss)
>
> > Discussion of stability issues in SoftKI and other low-rank GP approximations
>
> Since one apparent deficiency of the previous version of the draft was a lack of clarity motivating the deployment of the hutchinson pseudoloss we have deprecated the PCA comparison in favor of a numerical demonstration of the need for the hybrid pseudoloss protocol described in Section 3.2. This new change can now be found in section 4.3 titled "Numerical Stability" where a demonstration of the Exact MLL objective failing is given on the UCI dataset. Additionally, we have provided additional commentary to clarify some of the ambiguity of what we originally meant by "numerical instability" to begin with.
>
> > Explicit text or table detailing SoftKI's scaling/computational complexity
>
> This valuable idea has been added to section 2.4 and integrated into the writing accordingly.
>
> > In addition, I would like to see whether the numerical stability methods proposed by the authors provide benefit to other GP approximations (if applicable).
>
> We have added a new section to appendix B responding to this question we investigate the effect of using the QR stabilized linear solve procedure in SGPR. in general this correction does help SGPR's final performance, although the improvement is not very significant.